



# Controls on surface aerosol number concentrations and aerosol-limited cloud regimes over the central Greenland Ice Sheet.

Heather Guy[1,2], Ian M. Brooks[2], Ken S. Carslaw[2], Benjamin J. Murray[2], Von P. Walden[3], Matthew D. Shupe[4,5], Claire Pettersen[6], David D. Turner[8], Christopher J. Cox[5], William D. Neff[4,5], Ralf Bennartz[6,7], and Ryan R. Neely III[1,2]

[1]National Centre for Atmospheric Science, Leeds, U.K.
[2]School of Earth and Environment, University of Leeds, Leeds, U.K.
[3]Department of Civil and Environmental Engineering, Laboratory for Atmospheric Research, Washington State University, Pullman, WA, USA
[4]Cooperative Institute for Research in Environmental Sciences, University of Colorado, Boulder, CO, USA
[5]Physical Science Laboratory, National Oceanic and Atmospheric Administration, Boulder, CO, USA
[6]Space Science and Engineering Center, UW-Madison, Madison, WI, USA
[7]Earth and Environmental Sciences, Vanderbilt University, Nashville, TN, USA
[8]Global Systems Laboratory, National Oceanic and Atmospheric Administration, Boulder, CO, USA

**Correspondence:** Heather Guy (heather.guy@ncas.ac.uk)

**Abstract.** This study presents the first full annual cycle (2019-2020) of ambient surface aerosol number concentration (condensation nuclei > 20 nm, $N_{20}$) measurements collected at Summit Station, in the centre of the Greenland Ice Sheet (72.58° N, -38.45° E, 3250 m asl). The mean surface $N_{20}$ concentration in 2019 was 129 cm$^{-3}$, with the 6 h mean ranging between 1 cm$^{-3}$ and 1441 cm$^{-3}$. The highest monthly mean concentrations occurred during the late spring and summer, with the minimum concentrations occurring in February (mean: 18 cm$^{-3}$), demonstrating an opposite seasonal cycle in aerosol concentrations compared to low altitude Arctic stations (with latitudes > 66°N). High aerosol concentration events are linked to anomalous anticyclonic circulation over Greenland and the descent of free tropospheric aerosol down to the surface, whereas low aerosol concentration events are linked to cyclonic circulation over south-east Greenland that drives upslope flow and enhances precipitation en-route to Summit. Fog strongly effects $N_{20}$ concentrations, on average reducing $N_{20}$ by 20% during the first three hours of fog formation. Extremely low $N_{20}$ concentrations (< 10 cm$^{-3}$) occur in all seasons, and we suggest that fog, and potentially cloud formation, can be limited by low aerosol concentrations over central Greenland.

## 1 Introduction

The Greenland Ice Sheet (GrIS) has been losing mass at an unprecedented and accelerating rate since the early 21st century (Rignot et al., 2008, 2011; Van Den Broeke et al., 2016; Fettweis et al., 2017; Trusel et al., 2018; Shepherd et al., 2020), and as a result, has become the largest single contributor to global sea level rise (Van Den Broeke et al., 2016; Bamber et al., 2018; Slater et al., 2020). The majority of this mass loss is due to changes in the ice sheet surface mass balance (Slater et al., 2020), and in particular, increased surface melt and run off (Enderlin et al., 2014; Van Den Broeke et al., 2016; Shepherd et al., 2020). Clouds play a critical role in the ice sheet surface mass balance, both by providing mass input in the form of precipitation, and





by modulating the net radiation at the surface, thus influencing surface melt and run-off (Bennartz et al., 2013; Van Tricht et al.,
2016; Hofer et al., 2017; Miller et al., 2017). To make accurate projections of the future contribution of the GrIS to sea level
rise, models must correctly represent the properties of clouds and their interaction with the surface energy budget. Although
circulation anomalies drive a larger proportion of surface melt, discrepancies in cloud microphysical properties between dif-
ferent models currently result in larger uncertainties in future GrIS melt projections than the difference between low and high
greenhouse emission scenarios (Hofer et al., 2019). Amongst the largest uncertainties in cloud microphysical modelling are
the type, concentration and sources of aerosols (e.g. Seinfeld et al., 2016). Improving our understanding of aerosols and their
relationship with cloud properties over the GrIS is therefore key to reducing the uncertainty in future projections of GrIS melt
and global sea level rise.

Cloud properties are sensitive to the type and concentration of tropospheric aerosols (e.g. Twomey, 1977; Curry et al., 1996;
Storelvmo, 2017). Mixed-phase clouds in particular, which contribute significantly to surface warming over the GrIS (Miller
et al., 2015; Van Tricht et al., 2016), are sensitive to the number concentration of cloud condensation nuclei and ice-nucleating
particles (e.g. Norgren et al., 2018; Solomon et al., 2018); where cloud condensation nuclei (CCN) are a subset of aerosol
particles on which liquid droplets can form, and ice-nucleating particles (INP) are a subset of aerosols that can catalyze the
formation of ice crystals.

In ice-covered polar regions, CCN concentrations can be very low; surface CCN concentrations at 0.2% supersaturation
are usually less than 100 cm$^{-3}$ and can regularly fall below 10 cm$^{-3}$ in the high Arctic (e.g. Mauritsen et al., 2011; Leck
and Svensson, 2015), compared to typical values of over 1000 cm$^{-3}$ at rural mid-latitude sites (e.g. Schmale et al., 2018). In
cases where CCN are extremely low (< 10 cm$^{-3}$), the small number of sites for droplet activation limits cloud droplet number
concentration, and high supersaturations cause all available CCN to activate and grow to relatively large sizes, facilitating
further growth by collision and coalescence and resulting in precipitation as drizzle (Mauritsen et al., 2011). This generates
a positive feedback where the lack of CCN can result in total dissipation of the cloud (Mauritsen et al., 2011; Stevens et al.,
2018). Thus, within this CCN-limited regime, the availability of CCN becomes a dominant control on cloud formation and
longevity such that a small increase in concentration can lead to a decrease in droplet size that serves to reduce precipitation
efficiency, leading to a relative increase in cloud liquid water path (LWP) (Mauritsen et al., 2011). The change in LWP in turn
modulates the cloud longwave radiative effect (Mauritsen et al., 2011; Miller et al., 2015). Alternatively, the addition of CCN
when a cloud is not in the CCN-limited regime can have a cooling effect at the surface in the summer due to the associated
increase in cloud reflectivity of incoming solar radiation (Twomey, 1977; Intrieri et al., 2002). For optically thin clouds (< 40 g
m$^{-2}$), that are common at Summit (Shupe et al., 2013b; Miller et al., 2015), the smaller droplet size associated with increased
CCN results in higher cloud emissivity, increasing the downwelling longwave radiative flux and having a relative warming
effect at the surface (Lubin and Vogelmann, 2006; Garrett and Zhao, 2006). Understanding when and where each of these
processes dominate is extremely important for understanding cloud radiative forcing and the surface energy budget (Schmale
et al., 2021).

The concentration of ice-nucleating particles (INP) is also an important control on the longevity and radiative impact of
clouds. INP are required to form primary ice in supercooled liquid clouds that are warmer than the homogeneous freezing



temperature (approximately -38 °C) (e.g. Kanji et al., 2017). Because the low-level clouds that have the largest radiative effect
at the Arctic surface usually have temperatures between -38 °C and 0 °C (Shupe and Intrieri, 2004; Shupe et al., 2013b;
Miller et al., 2015), INP concentrations are an important control on the ice and liquid water contents of these clouds. Clouds
containing ice crystals are optically thinner than those containing only supercooled water droplets, and therefore emit less
longwave radiation towards the surface, having a relative cooling effect (e.g. Prenni et al., 2007). Even more importantly, once
ice crystals are present in a supercooled cloud, the lower saturation vapor pressure of ice versus liquid water results in the
preferential growth of ice crystals at the expense of liquid droplets. This is known as the Wegener-Bergeron-Findeisen (WBF)
process, the result of which is a decrease in LWP as droplets evaporate and an increase in precipitation due to the relatively
large ice crystals, ultimately leading to cloud dissipation (e.g. Lohmann and Feichter, 2005). Only a small percentage of CCN
can act as INP, and from what limited measurements exist, INP concentrations are also particularly low in the Arctic ($\sim 10^{-7}$
to $10^{-5}$ cm$^{-3}$, Wex et al., 2019). The lack of INP in the Arctic may contribute to the unusual persistence of low-level mixed-
phase stratoculmulus clouds (Morrison et al., 2012), that are highly important for radiative forcing at the surface, and played a
role in the anomalous GrIS surface melt event in 2012 (Bennartz et al., 2013).

Both CCN and INP concentrations are also important for precipitation accumulation. In liquid clouds, the increase in cloud
droplet number concentration and associated decrease in cloud droplet size under high CCN concentrations reduces the oppor-
tunities for droplet collision and coalescence and hence reduces precipitation relative to equivalent situations with lower droplet
concentrations (e.g. Lohmann and Feichter, 2005). In mixed-phase clouds this process is more complex, since changes in the
cloud droplet size distribution can have both positive and negative effects on the efficiency of ice production (Cheng et al.,
2010; Lance et al., 2011; Possner et al., 2017). Cloud phase partitioning is also important since ice phase clouds have markedly
different precipitation characteristics to those containing super-cooled liquid water (Pettersen et al., 2018; McIlhattan et al.,
2020). Model simulations generally overestimate precipitation accumulation over the GrIS (McIlhattan et al., 2017; Kay et al.,
2018; Lenaerts et al., 2020), and in particular the contribution from mixed-phase clouds. McIlhattan et al. (2017) find that the
CESM model overestimates snow frequency from mixed-phase clouds by 52% and underestimates the occurrence frequency
of liquid-bearing clouds by 21% over the central GrIS. This is consistent with an overly active WBF process in the model; a
process that is strongly controlled by INP concentrations.

To date, all observations of the CCN-limited regime (Mauritsen et al., 2011; Leaitch et al., 2016), and INP concentrations
(Wex et al., 2019), in the Arctic are located at marine or coastal sites. However, the central GrIS is a distinct Arctic environment
due to its high elevation (3250 m asl at its highest point, Fig. 1) and persistent ice cover (1.7 x $10^6$ km$^2$) that results in a year-
round high surface albedo. There are no substantial local sources of aerosol from the surface for over 400 km in any direction
from the center of the ice sheet. The year-round high surface albedo of the central GrIS (Box et al., 2012) results in unique
seasonality in cloud radiative forcing. Most parts of the Arctic have less snow and ice cover in the summer and hence a lower
albedo, during this time clouds can have a net cooling effect at the surface due their relatively high albedo (e.g. Shupe and
Intrieri, 2004). In contrast, over the central GrIS the seasonal change in surface albedo is negligible and clouds have a net
warming effect at the surface year-round (Miller et al., 2015; Van Tricht et al., 2016).





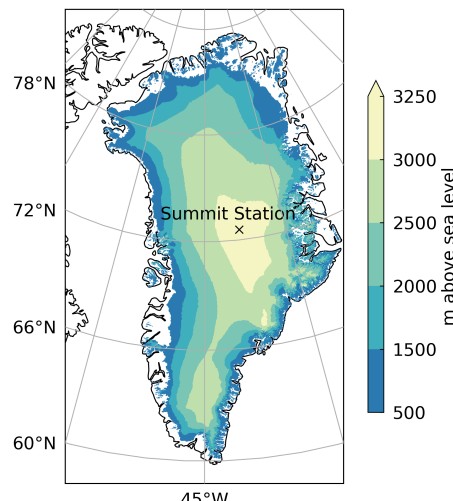

**Figure 1.** Location of Summit Station at the highest point on the Greenland Ice Sheet. Ice elevation contours are from the Greenland Ice Mapping Project (Howat et al., 2017).

The high elevation and extreme radiative cooling from the center of the GrIS drives low-level katabatic winds that radiate towards the ice sheet edge and, combined with synoptic and large scale circulation patterns, support the formation of a persistent high pressure system over Greenland (Heinemann and Klein, 2002; Hanna et al., 2016). For this reason, Greenland has been referred to as the 'northern wind pole', where upper level air currents driven by the Hadley circulation descend and return to lower latitudes (Hobbs, 1945; Heinemann and Klein, 2002). The descent of upper tropospheric air to the surface of the central GrIS results in a larger contribution of well-mixed free tropospheric aerosol (Stohl, 2006). Hence the transport processes and source regions controlling the concentrations of aerosols over the central GrIS are distinct from other Arctic sites (Hirdman et al., 2009; Backman et al., 2021).

The presumed insignificant local aerosol sources at the surface of the GrIS suggests that both low CCN concentrations with the potential to limit cloud formation, and low INP concentrations that can control cloud phase, could certainly occur. The difference in aerosol transport pathways to the GrIS when compared to coastal or marine Arctic sites implies that the processes controlling aerosol-limited cloud regimes, and their frequency of occurrence, might differ substantially from other Arctic locations. Hence, a thorough analysis of the role of the aerosol-limited conditions over the GrIS is warranted, especially given the unique sensitivity of the GrIS to longwave cloud forcing.

Despite the potential for aerosol-limited clouds to effect the surface mass balance of the GrIS, and the large uncertainties in modelled cloud microphysical properties over Greenland (Hofer et al., 2019; Schmale et al., 2021), there are very few observations of aerosol number concentration over the central GrIS, and those that do exist are mostly limited to the summer





season (Flyger et al., 1976; Hogan et al., 1984; Davidson et al., 1993; Bergin et al., 1994, 1995). This study presents the first full year of surface aerosol total number concentration measurements from Summit Station, in the central GrIS, which can be used as a baseline for future modelling studies investigating the effect of cloud-aerosol interactions on the GrIS surface energy budget and mass balance. We assess local and synoptic controls on surface aerosol concentrations at Summit and present three case studies where extremely low total aerosol number concentrations ($< 10\ \mathrm{cm}^{-3}$) coincide with cloud dissipation, indicating
that CCN-limited clouds occur over the central GrIS and could be an important contributor to the surface energy budget.

## 2 Measurements and methods

All observations in this study were made at Summit Station (Summit), a scientific research base funded by the U.S. National Science Foundation. Summit is located at the highest point on the GrIS (3250 m asl) and is over 400 km from the coast in the east and west directions, and over 1,000 km from the south west and south east coasts (Fig. 1). Aerosol, cloud and
atmospheric profile measurements were collected as part of the ICECAPS-ACE project: ICECAPS (Integrated Characterisation of Energy, Clouds and Precipitation at Summit) has been operating at Summit since 2010 and consists of a suite of ground based remote sensing instrumentation and twice-daily radiosonde launches (Shupe et al., 2013b). The ACE (Aerosol Cloud Experiment) addition to ICECAPS began collecting data in February 2019 and includes measurements of surface aerosol number concentration and size distribution in addition to turbulent and radiative fluxes used to characterize the surface energy
budget. This study uses a subset of ICECAPS-ACE data listed in Table 1, as well as meteorological measurements from the NOAA Global Monitoring Laboratory (GMLMET, 2021). The references in Table 1 provide additional information on the instruments and methodologies for the derived parameters. Section 2.1 provides the details of the aerosol number concentration sampling and quality control.

To investigate the effect of near-surface local processes that have the potential to modify surface aerosol concentrations, we
look at four event types: fog, precipitation, blowing snow (BLSN), and strong surface-based temperature inversions (SBIs). For each type, we examine the change in surface aerosol concentrations across multiple events. To qualify, events of each type must last at least 60 minutes and separate events of the same type must be at least 5 hours apart. Subsections 2.2 to 2.4 provide specific details about how each event type is defined.

To assess the synoptic controls on surface aerosol concentrations, we use ERA5 reanalysis data (Hersbach et al., 2020)
made available by the European Centre for Medium-Range Weather Forecasts (ECMWF). ERA5 is the highest resolution global reanalysis product to date, with ~15 km horizontal resolution over Greenland, 137 pressure levels up to 80 km, and 1 h temporal resolution. We also use ERA5 reanalysis to drive the FLEXPART Lagrangian particle dispersion model (Pisso et al., 2019) to simulate aerosol transport pathways and surface emission sensitivities. Section 2.5 provides further details about the FLEXPART experimental design.





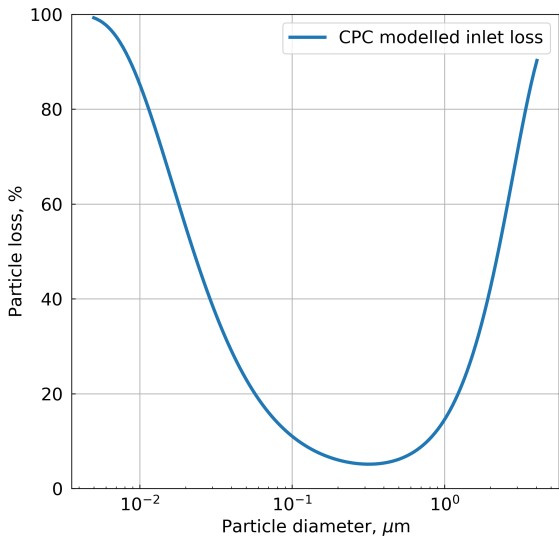

**Figure 2.** Modelled particle loss as a function of particle diameter in the CPC inlet, as estimated by the Particle Loss Calculator (Von Der Weiden et al., 2009)

## 2.1 Surface aerosol number concentrations

A condensation particle counter (GRIMM CPC 5.400) measured the ambient number concentration of condensation nuclei at 1 Hz frequency. The omni-directional conical inlet head was located ~3 m above the surface (this varied slightly throughout the observation period with snow drifting and accumulation) and air was sampled with a flow rate of 0.3 L min$^{-1}$. The inlet was connected to the CPC via a 6 m length of conductive silicone tubing with an 8 mm inner diameter. Although the CPC is calibrated to measure condensation nuclei > 5 nm diameter, the addition of the long inlet results in a loss of particles inside the tubing. Figure 2 shows an estimation of the loss of aerosol particles inside the inlet generated by the Particle Loss Calculator (Von Der Weiden et al., 2009). Smaller particles are increasingly lost due to diffusion to the walls of the inlet, and larger particles due to sedimentation and deposition. The Particle Loss Calculator does not account for the temperature gradient within the tubing, however, because the cold air in the inlet stream transitions into a warmer inlet (inside the heated building), this will act to reduce the loss of particles (Von Der Weiden et al., 2009). Also, because aerosol concentrations are small (<< 100,000 cm$^{-3}$), particle loss due to coagulation is negligible (Von Der Weiden et al., 2009). Based on these modelled inlet losses, the CPC measured condensation nuclei with diameters between 20 nm and 2.3 $\mu$m with over 50% efficiency (Fig 2). For this reason we henceforth refer to the CPC concentration measurements as $N_{20}$, indicating number concentrations of particles with diameter > 20 nm. Modelled inlet losses are < 15 % for particles with diameters between 0.08 and 1 $\mu$m, which is representative of the typical size range of CCN in clean Arctic environments (Hudson and Da, 1996; Leaitch et al., 2016).





To filter out data that may have been impacted by local station pollution, we omitted measurements collected when wind speeds are $< 1$ m s$^{-1}$ and when the wind direction is such that contaminated air from station operations may have advected across the inlet (between $270°$ and $360°$ from true north). A comparison between two OPC-N3 optical particle counters (described further in section 2.2), located at the opposite sides of camp, confirmed that these criteria are sufficient to account for

the impact of local station pollution (not shown). The removal of data associated with particular surface wind conditions may bias the dataset, however during the measurement period considered in this study wind speeds $< 1$ m s$^{-1}$ only occur 3.4 % of the time and polluting wind directions only occur 9.1 % of the time.

## 2.2 Detection of fog

Supercooled liquid fog is common at Summit and occurs in all seasons, with a minimum occurrence in April and maximum in

September (Cox et al., 2019). Fog droplets form on CCN and grow by condensation to typical diameters of 15 to 25 $\mu$m (Cox et al., 2019). Particles larger than $\sim 3$ $\mu$m cannot pass through the CPC inlet (Fig. 2), hence during fog events, the CPC measures the interstitial aerosol concentration. In this way, fog can result in extremely low surface aerosol concentration measurements that are not representative of the aerosol population outside of the fog (Bergin et al., 1995). In the absence of an instrument designed specifically to detect fog at Summit, we use data from an Alphasense Optical Particle Counter (OPC-N3, Crilley

et al., 2018) located next to the CPC inlet to identify fog periods.

The OPC-N3 resolves particle size distribution in 24 bins between 0.35 $\mu$m and 40 $\mu$m diameter. Natural aerosols with diameters greater than 10 $\mu$m are highly unlikely to be present in central Greenland due to the large distance from the source of any coarse mode aerosols and the large dry deposition velocity of such particles (Giorgi, 1986; Jaenicke, 1990). Under this assumption, particles detected by the OPC-N3 with diameters over 10 $\mu$m must be fog droplets. Real-time data monitoring at

Summit and comparison with visual observations for 6 months confirm that the OPC-N3 detects particles within this size range during both fog and blowing snow. At Summit, 80% of cases of drifting or blowing snow reported by onsite observers in 2019 occurred when the 3h mean 10 m wind speed was $> 6$ m s$^{-1}$; we remove all cases with wind speeds above this threshold to separate fog events from possible blowing snow events. We classify fog events as when the total concentration of particles with diameters $> 10$ $\mu$m is greater than 0.1 cm$^{-3}$. Figure 3 provides an example of the detection of fog using this methodology and

the associated reduction in $N_{20}$ concentration measured by the CPC.

Comparing this OPC-N3 fog classification to manual onsite observations reported at 00Z, 12Z and 18Z daily, the OPC-N3 does not detect fog when fog is reported by the observer (false negatives) in 35 out of 152 cases (23%). Six of these cases can be attributed to inconsistent observer log entries or to logged issues with the OPC-N3, some others may result from discrepancies between the actual and reported observation time. However, false positive detection is rare, occurring in only 6 cases (1%).

Therefore, although some fog events might be missed by the OPC-N3 fog classification, it is an accurate indicator of fog presence, with on-site observers confirming the presence of fog 99% of the time. The OPC-N3 was in operation between June and December 2019, and during this time the data are 96% complete.





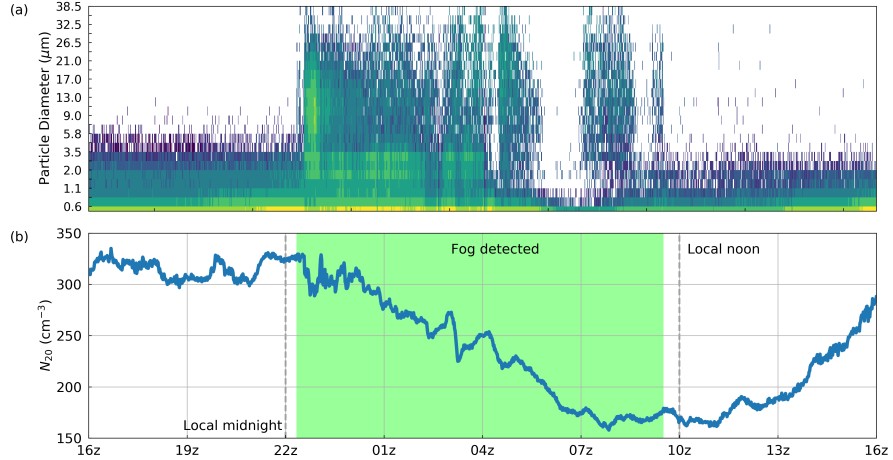

**Figure 3.** (a) Aerosol size distribution from the OPC-N3 from 31 July 2019 1600Z to 01 August 2019 1600Z (1 min averages). (b) $N_{20}$ aerosol number concentration from the CPC during the same period (1 min averages). The duration of the fog event identified by the methodology described in Section 2.2 is shaded in green.

## 2.3 Detection of precipitation and blowing snow

Below-cloud wet scavenging during snowfall can also reduce surface aerosol concentrations (e.g. Martin et al., 1980; Para-
monov et al., 2011). A Precipitation Occurrence Sensor System (POSS) (Sheppard and Joe, 2008) located about 2 m above ground level at Summit measures the Doppler velocity spectrum of hydrometeors within a 1 m$^3$ sampling volume. Surface snowfall rate retrieved from the POSS agrees well with retrievals from the lowest reliable range gate of the Millimeter Cloud Radar (MMCR) at Summit, with a root mean squared error of 0.08 mm h$^{-1}$ (Castellani et al., 2015). The 'POSS power unit' (the zeroth moment of the Doppler spectrum) can be used as a binary indicator of precipitation, and in this study we use a
threshold of 2 POSS power units to identify precipitation events and exclude blowing snow, as per Pettersen et al. (2018). POSS data are 95% complete between June and December 2019.

The wind speed threshold for blowing snow (BLSN) varies depending on temperature and the properties of surface snow (Schmidt, 1982; Mann et al., 2000). In section 2.2 we used a 6 m s$^{-1}$ threshold as a minimum to avoid cases of possible blowing snow. However, to positively identify BLSN events we use a 10 m wind speed threshold of $\geq 9$ m s$^{-1}$. During 2019,
on-site observers reported blowing or drifting snow 99% of the time when the 3 h mean was above this threshold.

## 2.4 Detection of surface-based temperature inversions

surface-based temperature inversions (SBIs) occur at Summit in all seasons due to strong and persistent radiative cooling of the surface. SBIs are most common in the winter (Oct-Mar) where they occur over 70% of the time with a typical magnitude of ~5 °C between 10 m and 2 m above the surface (Miller et al., 2013). In the summer (JJA), the amplitude of SBIs is weaker, and



they only occur ∼30% of the time (Miller et al., 2013). The presence of a SBI limits the turbulent mixing of air (and aerosols) down to the surface. In this case, aerosol concentrations measured at the surface may not be representative of concentrations at cloud level (Igel et al., 2017), as scavenging and dry deposition in the stable layer might result in extremely low surface aerosol concentrations (Dibb et al., 1992). To explore the effect of SBIs on surface aerosol concentrations in this study we classify SBI events where the 15 m minus the 2 m (above ground level) temperature difference must be greater than 3 °C (> 0.23 °C m$^{-1}$).

Detection of SBI events is limited to June through October 2019 due to outages in the 15 m temperature sensor, but during this time data were 97% complete.

## 2.5 Aerosol source regions and transport pathways

The FLEXPART Lagrangian particle dispersion model (Pisso et al., 2019) is used to simulate aerosol transport pathways and surface emission sensitivities throughout 2019. FLEXPART simulations are run every 6 hours and driven by reanalysis data

from ERA5, at the same horizontal and vertical resolution as the input data. In each simulation, 40,000 'particles' are released at 100 m above the surface at Summit, and FLEXPART traces each particle back in time for 20 days. Particles follow the mean 3D wind field from ERA5 combined with a stochastic 3D turbulence field and parameterized convection (Forster et al., 2007). FLEXPART also simulates wet and dry deposition as linear decay constants based on a user input particle mean diameter, density, water and ice nucleation efficiency. In both cases deposition acts to reduce the total 'mass' of each particle, and a

particle's back trajectory stops when its mass reaches zero. Due to limited prior information about aerosols at Summit, we used the default aerosol tracer species; which assumes a particle mean diameter of 2.5 μm, density of 1400 kg m$^{-3}$, and water and ice nucleation efficiencies of 0.9 and 0.1 respectively. Although the mean particle diameter at Summit is likely smaller, particles of 2.5 μm diameter are observed (VanCuren et al., 2012), and their relatively long atmospheric lifetimes ( > 10 days in the middle-upper troposphere; Jaenicke, 1990) enable them to advect over the GrIS, therefore they ought to be a good

indicator of possible aerosol source regions. Further, simulation tests (not shown) showed that varying these parameters results in changes in the absolute values of emission sensitivity (depending on the relative efficiencies of wet and dry deposition) but does not noticeably modify the spatial distribution of surface emission sensitivity, so our inference of possible aerosol source regions is not sensitive to this choice. FLEXPART outputs gridded emission sensitivity and supplementary back trajectory data that include the mean (centroid) back trajectory of all particles for each simulation, and the percentage of particles within

the planetary boundary layer (PBL) at each time step. The surface emission sensitivity is proportional to the total amount of time that all particle back trajectories have spent near the surface (0-2,000 m) during the simulation period, representing the probability that aerosols emitted from each grid cell would have been detected at Summit at the simulation start time. We plot surface emission sensitivity as a percentage of the maximum value, to facilitate comparisons between figures.


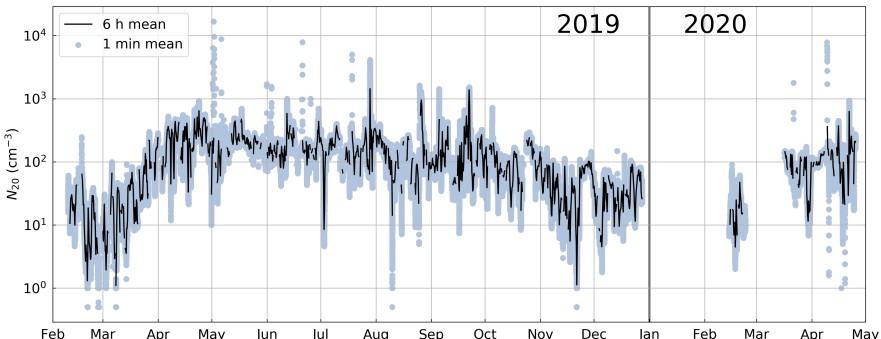

**Figure 4.** Surface $N_{20}$ concentrations from the CPC at Summit from February 2019 until May 2020.

## 3 Results

### 3.1 Surface aerosol number concentrations at Summit, 2019-2020

The mean surface $N_{20}$ concentration in 2019 was 129 cm$^{-3}$, with the 6 h mean ranging between 1 cm$^{-3}$ and 1441 cm$^{-3}$ (Fig. 4). These results are of the same order of magnitude as previous summertime measurements of condensation nuclei at Summit (100-500 cm$^{-3}$ in the first week of July 1992, Bergin et al., 1994), and from DYE III on the south east GrIS ($\sim$ 6-1000 cm$^{-3}$ in July and August 1982, Hogan et al., 1984). These results are also comparable in magnitude to $N_{10}$ concentrations measured at other Arctic stations ($\sim$ 1-2000 cm$^{-3}$ at Utqiagvik, Alaska, and $\sim$ 5-3000 cm$^{-3}$ at Pallas, Finland; Asmi et al., 2013), however the seasonal cycle is notably different (see section 4.1). The minimum $N_{20}$ concentrations in 2019 at Summit occur in late February and early March, followed by a sharp increase of 2 orders of magnitude throughout March and April (Fig. 4). Between May and October, concentrations are fairly consistent and on the order of 100 cm$^{-3}$ before decreasing again between October and December. Although data in early 2020 is limited, a similar increase in concentrations between February and May is apparent (Fig. 4).

### 3.2 The effect of local surface processes on aerosol concentrations

Surface aerosol concentration is not necessarily representative of concentrations at cloud level (Igel et al., 2017; Creamean et al., 2021). Assuming that there are no significant local surface sources at Summit, surface aerosol concentrations are unlikely to be higher than concentrations at cloud level, but boundary layer processes that result in a higher rate of deposition at the surface than replenishment from above could result in significantly lower aerosol concentrations at the surface. Notably, this can occur in the event of surface fog (Bergin et al., 1995) or surface temperature inversions that inhibit turbulent mixing (Dibb et al., 1992). Precipitation and blowing snow can also effect surface aerosol concentrations, although Bergin et al. (1995) did not observe a significant effect of precipitation on surface aerosol concentration at Summit. The effect of blowing snow is uncertain because it can act to scavenge aerosol through impaction, but can also re-release deposited aerosol into the




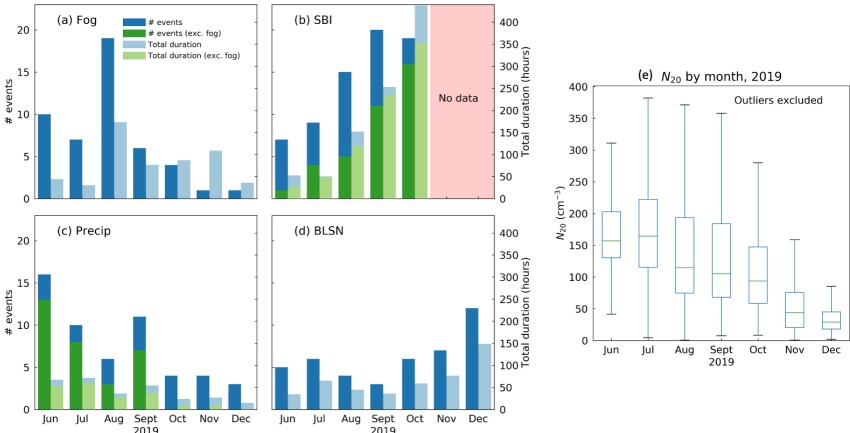

**Figure 5.** Frequency and duration of (a) fog events, (b) surface-based temperature inversion events, (c) precipitation events and (d) blowing snow events, detected between June and December 2019 using the methodology described in Sections 2.2-2.4. Blue bars include all events and green bars show the change in distribution for SBI and precip events after the removal of events containing fog. (e) The distribution of $N_{20}$ for the same months, excluding outliers.

boundary layer through sublimation (e.g. Frey et al., 2020). Here we assess the effect of these local surface processes on $N_{20}$ concentrations at Summit between June and December 2019 (whilst the OPC-N3 was operational).

The OPC-N3 identified 48 distinct fog events during this period. The longest cumulative fog duration was in August (Fig. 5a) when fog was present for $\sim 23\%$ of the month, consistent with previous multi year observations of supercooled liquid fogs at Summit (Cox et al., 2019). The mean duration of fog events was 3.3 hours and the longest event lasted 9.8 hours.

SBI events were also present in all months, and increased in total duration from summer to winter (Fig. 5b), again consistent with previous observations (Miller et al., 2013). The average duration of SBI events was 8.4 hours and the longest individual event lasted 5.8 days. SBI and fog events are not independent since fog condensate often forms due to surface cooling associated with the establishment of SBIs (e.g. Cox et al., 2019). Just under half all detected SBI events also contained fog (Fig. 5b), although because fog events are typically shorter, this only accounted for 17% of the total SBI duration.

Precipitation frequency and duration was highest in the summer and lowest in November and December (Fig. 5c). The average duration of precipitation events was 2.9 hours and the longest event lasted 14.1 hours. In contrast, BLSN events occurred most frequently in November and December, with an average duration of 6.9 hours (Fig. 5d). The seasonal distribution and duration of precipitation and BLSN events are also consistent with previous multi-year observations (Castellani et al., 2015; Pettersen et al., 2018; Bennartz et al., 2019; Cox et al., 2019). Fog was detected during 23 of the 54 precipitation events (Fig.

5c). Because the OPC-N3 does not distinguish between fog and BLSN, it is not possible to determine how often fog might have been present during BLSN events.

Figure 6 shows the median change in surface $N_{20}$ concentration during the first 3 hours of each event type. Only during fog events is there a consistent change: After three hours, the majority of fog events show a reduction in surface $N_{20}$ concentration

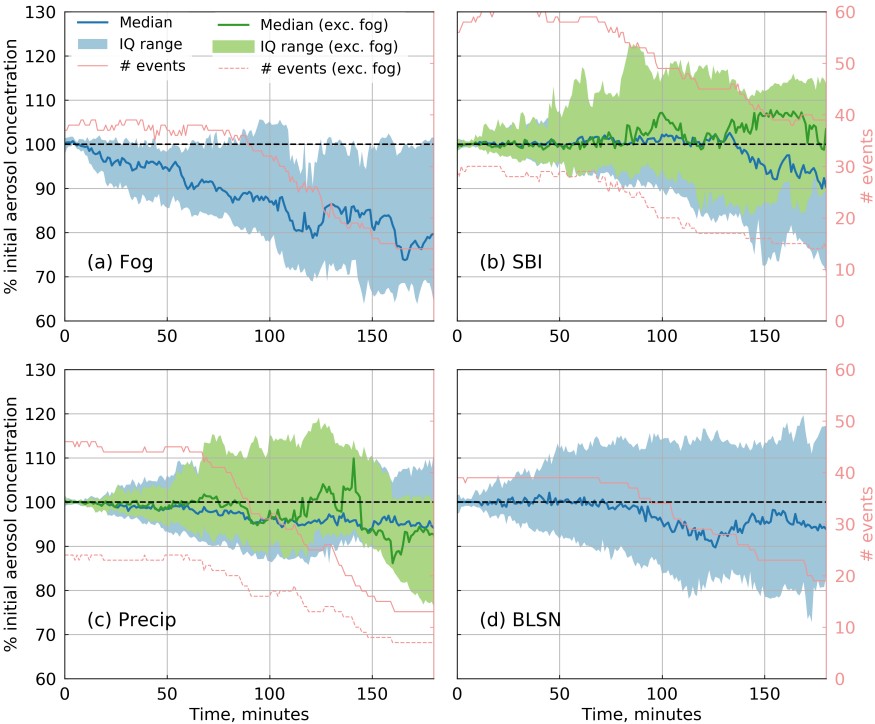

**Figure 6.** The change in surface aerosol concentration (%) over time during the first 3 h of each event for (a) fog, (b) surface-based temperature inversions, (c) precipitation and (d) blowing snow events. The thick blue line and blue shading are the median and interquartile range of all events, the thick green line and green shading are the median and interquartile range for SBI and precip events that do not contain fog. The pink line indicates the total number of events at each time step for all events (solid) and excluding fog events (dashed).

by up to 35% (Fig. 6a). For SBI events, there is very little discernible change in surface $N_{20}$ concentrations during the first
two hours (Fig. 6b). After $\sim 140$ min there is a small median reduction in $N_{20}$ concentration that is not present when events
that contain fog are omitted. This implies the formation of a SBI alone is not a strong control on surface aerosol concentrations
at Summit in the initial three hours of the event. During both precipitation and BLSN events, the median change in surface
aerosol concentrations remains close to zero (Fig. 6c, d).

## 3.3   Synoptic controls on surface aerosol concentrations

Here we explore the general relationship between surface $N_{20}$ concentrations and synoptic conditions during 2019. Because
both $N_{20}$ concentrations and variables that change on synoptic timescales (i.e. surface pressure, geopotential height) vary
seasonally, this seasonal dependence is removed prior to analysis. To calculate surface $N_{20}$ concentration anomalies we subtract
the monthly median value for 2019. For all other variables (from GML-MET, 2020, and ERA5) anomalies are calculated by
subtracting the 10 year (2009-2019) monthly mean climatology. Generally throughout 2019 anomalous changes in the 3 day


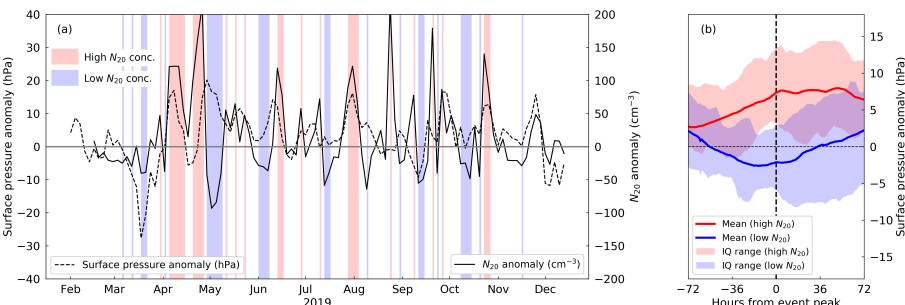

**Figure 7.** (a) Surface pressure anomaly (3 day mean, solid black line) and $N_{20}$ anomaly (3 day median, dashed) during 2019. Red and blue shading highlight high and low aerosol concentration events respectively (high/ *low* events are where the $N_{20}$ anomaly is above the 75th/ *below the 25th* percentile). (b) The mean and interquartile range in surface pressure anomaly across all high (red) and low (blue) $N_{20}$ events, for the 72 hours before and after the maximum *(minimum)* $N_{20}$ for each high *(low)* event.

mean surface pressure are in phase with anomalous 3 day median $N_{20}$ concentrations, with some exceptions (Fig. 7a). To look at typical synoptic conditions associated with anomalous $N_{20}$ concentrations at Summit, we look at 'high' and 'low' $N_{20}$ concentration events, where the 3 day median $N_{20}$ anomaly is greater than the 75th percentile or less than the 25th percentile respectively. To avoid oversampling, any events separated by less than four days are combined into a single event. The resulting high and low $N_{20}$ events are highlighted in Fig. 7a (15 high events and 14 low events).

On average, an increase in surface pressure anomaly precedes anomalously high $N_{20}$ events, with the maximum $N_{20}$ coinciding with surface pressure anomalies leveling off (Fig. 7b). In contrast, a decrease in surface pressure anomaly precedes the majority of low $N_{20}$ events, with the minimum $N_{20}$ coinciding with the minimum surface pressure anomaly on average (Fig. 7b). Averaged over all high $N_{20}$ concentration events, 500 hPa geopotential heights are anomalously high (by over 75 m in central Greenland) and there is an anomalous anti-cyclonic circulation over the GrIS (Fig. 8a). In contrast, when averaged over

the low $N_{20}$ concentration events, there is a region of anomalously low geopotential heights and cyclonic circulation centered on south-east Greenland (Fig. 8b).

    FLEXPART simulations of surface emission sensitivity during the high $N_{20}$ concentration events show that sensitivity to surface emissions in the 20 days prior to detection at Summit outside of the ice sheet itself is rare (Fig. 9a), although there is some sensitivity to emissions from North America and Europe. Because there are no significant aerosol sources over the

ice sheet itself, this implies that most particles arriving at Summit during these events have been high in the atmosphere ($>$2,000 m agl) for over 20 days prior to detection at Summit. This is supported by the low percentages of particles in the planetary boundary layer, and relatively high mean altitude of all particles during the high $N_{20}$ concentration events (Fig. 9a). In contrast, the surface emission sensitivity during the low $N_{20}$ concentration events covers a broader area, encompassing coastal Greenland, Iceland, the Canadian Arctic and the intervening north Atlantic Ocean (Fig. 9b). There is a much higher

percentage of particles in the boundary layer in the week preceding detection at Summit during the low $N_{20}$ events than during





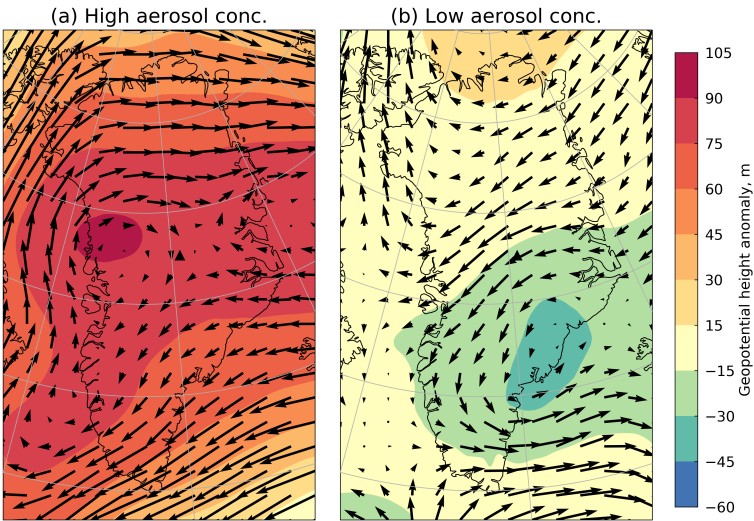

**Figure 8.** ERA5 mean 500 hPa geopotential height anomaly (shaded) and 500 hPa horizontal wind anomalies (barbed) for all high $N_{20}$ concentration events (a) and low $N_{20}$ concentration events (b).

the high $N_{20}$ events, and during the low $N_{20}$ events the particles are transported up to the highest point of the ice sheet from lower elevations (Fig. 9b).

### 3.4 Case studies of potential aerosol-limited cloud regimes at Summit

Fig. 4 demonstrates that surface $N_{20}$ concentrations fall below 10 cm$^{-3}$ in all seasons at Summit, suggesting that surface
CCN concentrations fall below this threshold even more frequently. Given the existing evidence that aerosol concentrations this low can limit cloud formation elsewhere in the Arctic (Mauritsen et al., 2011; Stevens et al., 2018), we hypothesize that fog formation can be limited by low CCN concentrations over central Greenland, and if there are occasions where the surface aerosol concentration is representative of concentration at cloud height, that cloud formation can be limited by low CCN concentrations too. In this section we look in detail at three events where extremely low aerosol concentrations ($N_{20} < 10$
cm$^{-3}$ for $> 3$ hours) coincided with cloud dissipation in the absence of fog, to look for further evidence of CCN-limited cloud regimes at Summit.

    For each of the three cases considered (3 July 2019, 10 August 2019, and 21 November 2019), air is advected to the top of the ice sheet from different directions: On 3 July 2019, the primary aerosol source region is northern Siberia (Fig. 10a), on 10 August 2019, air approaches Summit from the north via the Canadian Arctic Archipelago (Fig. 10b), and on 21 November
2019, air approaches Summit from the south east, and is sensitive to emissions from northern Quebec (Fig. 10c). Two of the





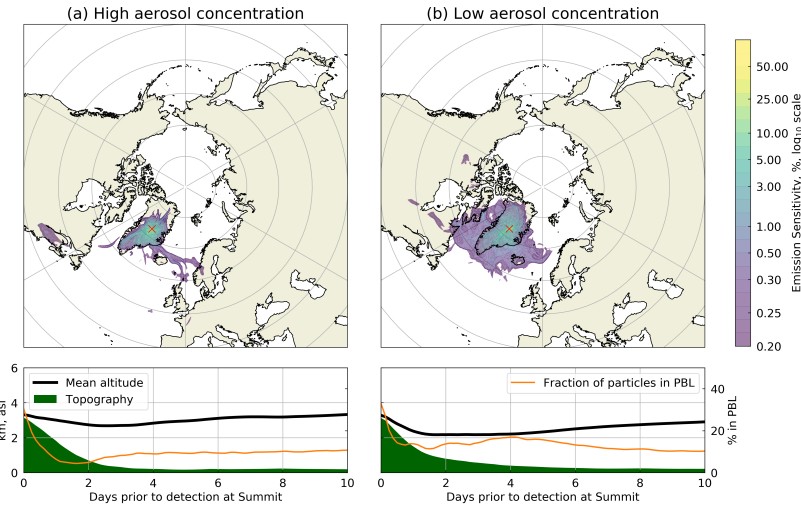

**Figure 9.** Results from FLEXPART back trajectory simulations averaged over the high $N_{20}$ concentration events (a) and the low $N_{20}$ concentration events (b). Upper: Surface emission sensitivity aggregated over 20 days prior to detection at Summit (as a percentage of the maximum value). Lower: Mean altitude and fraction of particles within the planetary boundary layer (PBL) for all simulated particles over the 10 days prior to detection.

three case studies (3 Jul 2019 and 21 November 2019) occur in the presence of anomalously low 500 hPa geopotential heights over south-east Greenland, with a stronger than usual south easterly wind component drawing air up the ice sheet from the south east coast (Fig. 10a and 10c). On both of these occasions > 50 % of particles are within the PBL 4-6 days prior to arrival at Summit. On the 10 August 2019 case, there is an anomalous region of high 500 hPa geopotential heights over north-west Greenland and a stronger than usual northerly wind component over Summit (Fig. 10b). Although the FLEXPART simulated particles remain closer to the ground for a longer period of time, the percentage of particles within the PBL in the 10 days prior to detection at Summit is much lower on 10 August 2019 than in the other two cases (Fig. 10b). On all three occasions, air is advected up to the ice sheet to Summit from lower elevations and spends > 1 day prior to detection at Summit within the lowest 800 m above the surface of Greenland. The local conditions associated with each case are outlined below.

### 3.4.1   03 July 2019

On 02 July 2019, surface aerosol concentrations dropped rapidly from $\sim 200$ cm$^{-3}$ to $< 10$ cm$^{-3}$ over a period of $\sim 9$ h (Fig. 11a). The 12Z equivalent potential temperature profile on 02 July 2019 shows that the lowest layer of broken stratocumulus cloud existed within a well mixed boundary layer (Fig. 11e). Shortly after 18Z, aerosol concentrations dropped below 10 cm$^{-3}$, and there was a reduction in cloud cover (Fig. 11a-c). The on-site observer log recorded a transition from broken altocumulus at 18Z to few clouds and unlimited visibility at 00Z on 03 July 2019, despite the fact that the lowest 200 m above the surface



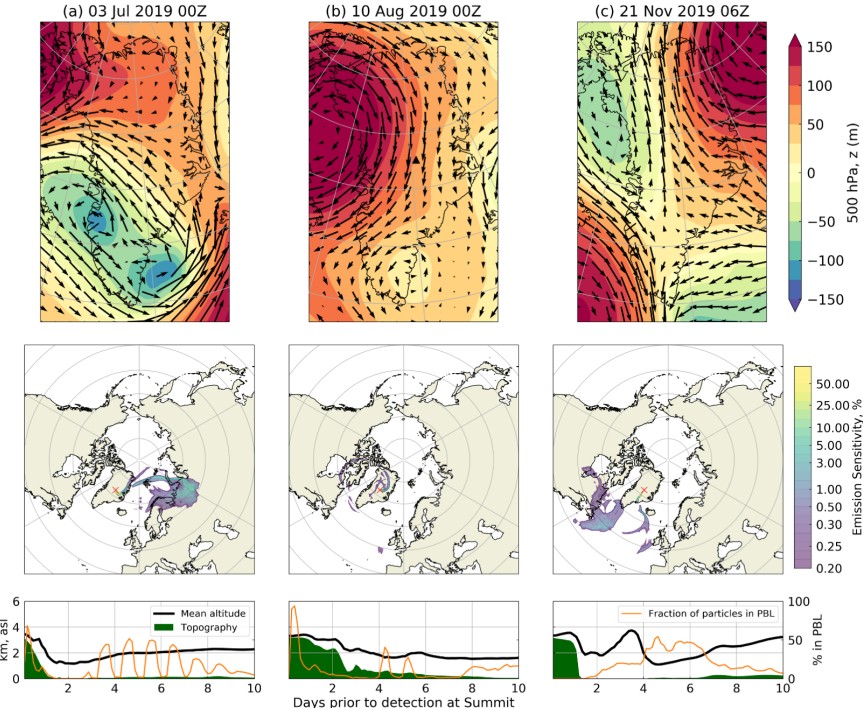

**Figure 10.** Synoptic anomaly plots and aerosol transport pathways during the three low aerosol cases studies. Upper row: 500 hPa geopotential height and horizontal wind anomalies from ERA-5. Middle row: FLEXPART surface emission sensitivity (as a percentage of the maximum value) over the 10 days prior to aerosol detection at Summit. Lower row: FLEXPART mean aerosol transport height (back bold line) and percentage of particles within the planetary boundary layer (orange line) over the 10 days prior to detection at Summit. The shaded green area represents the mean height of topography beneath all particles.

remained saturated with respect to water (Fig. 11f). On this occasion, the 00Z radiosonde was launched from the surface at 2315Z, and typically the weather observation is recorded at the time of launch. Photographs from a webcam viewing the aerosol inlet (taken every 15 minutes and orientated towards the eastern horizon) confirm that skies were clear and visibility was good at 2315Z, but by 2330Z there was a clearly visible fog bow, indicating liquid fog droplets (Fig. 12). Notably on this occasion,

aerosol concentrations fell to $< 10$ cm$^{-3}$ in the absence of fog. At 0015Z visibility was obscured and the OPC-N3 detected fog at the surface (Fig. 12). The fact that the OPC-N3 did not detect fog droplets until 0015Z, despite an increase in LWP at 2330Z (Fig. 11d), could be explained by either a) the droplets forming the fog bow at 2330Z were too large to be detected by the OPC-N3 ($> 40$ $\mu$m diameter) or b) the fog was in the process of descending to the surface. In either case, both a) and b)





support the hypothesis of Mauritsen et al. (2011), that in the absence of sufficient CCN, any existing CCN activate and grow
to relatively large sizes, falling to the surface as 'drizzle'.

The rapid transition from clear skies (despite a saturated surface layer) at 2315Z to fog at 0015Z coincided with aerosol
concentrations beginning to increase again (Fig. 11a). As aerosol concentrations continued to increase, a thin low-level mixed-
phase cloud returned and gradually lifted and thickened. By 12Z on 03 July 2019, aerosol concentrations had returned to $\sim 200$
cm$^{-3}$ and the lowest cloud layer had developed into a typical Arctic mixed-phase cloud (Shupe et al., 2006; Morrison et al.,
2012; Shupe et al., 2013b) with a cloud top close to 500 m (Fig 11c), capping a well-mixed boundary layer (Fig. 11e).

### 3.4.2   10 August 2019

At 12Z on 09 August 2019 there was a 2.5 km deep cloud over Summit (Fig. 13b) and surface $N_{20}$ concentrations were $\sim 100$
cm$^{-3}$ (Fig. 13a). Between 14Z and 18Z, there was a sharp decrease in liquid water path (Fig. 13d) as the cloud thinned until
there was nothing detected by the radar at 19Z (Fig. 13b). At this time, the surface-based temperature inversion strengthened
to $> 0.23$ °C m$^{-1}$ (Fig. 13a), likely due to the increase in longwave cooling at the surface after the reduction in cloud cover. In
this case, it was only after the strengthening of the surface temperature inversion that aerosol concentrations began to decrease.
At 2140Z the OPC-N3 detected fog, and aerosol concentrations decreased more rapidly, falling below 10 cm$^{-3}$ at 2335Z and
reaching a minimum of 0.5 cm$^{-3}$ at 01Z on 10 August 2019 after which the fog thinned and cleared (Fig. 13a). Aerosol
concentrations began to rise again from 0215Z, and when they increased above 10 cm$^{-3}$ at 0400Z, there was a sudden sharp
increase in liquid water path (Fig. 13d), and a thin low mixed-phase cloud developed (Fig. 13c). The cloud thickened as aerosol
concentrations continued to increase back to $\sim 100$ cm$^{-3}$ at 07Z. Fig. 14 shows the transition from cloudy to clear skies, then
to thin fog and back to overcast again throughout this event.

### 3.4.3   21 November 2019

Surface $N_{20}$ concentrations decreased from 50 cm$^{-3}$ at 06Z on 20 November 2019 to a minimum of 0.5 cm$^{-3}$ at 0630Z on
21 November 2019, and remained below 10 cm$^{-3}$ for a total of 24 hours (Fig. 15a). As aerosol concentrations decreased,
a low-level mixed-phase cloud thinned and liquid water path fell to 0 g m$^{-2}$ by 09Z (Fig. 15c,d). The 20 November 12Z
radiosonde shows that the boundary layer was neutrally stratified up to about 300 m, above a very shallow stable surface layer
(where the air temperature increased 7 °C in the 4 m immediately above the surface) (Fig. 15e). At 00Z on 21 November 2019
the temperature inversion in the lowest 4 m of the atmosphere strengthened to 12 °C, the sky above Summit was clear, and
$N_{20}$ concentrations continued to fall until 06Z. At 12Z on 21 November 2019 a 3 km deep ice cloud moved across Summit
(Fig. 15b,c) and $N_{20}$ concentrations began to increase again (Fig. 15a). Liquid water path initially remained close to zero but
increased sharply when $N_{20}$ concentrations rose above 10 cm$^{-3}$ at 00Z on 22 November 2019. Between the 20 November
2019 12Z and the 22 November 2019 00Z radiosonde profile, the 3 km agl potential temperature decreased by $> 5$ °C (Fig.
15e), possibly indicating an air mass transition during this period.



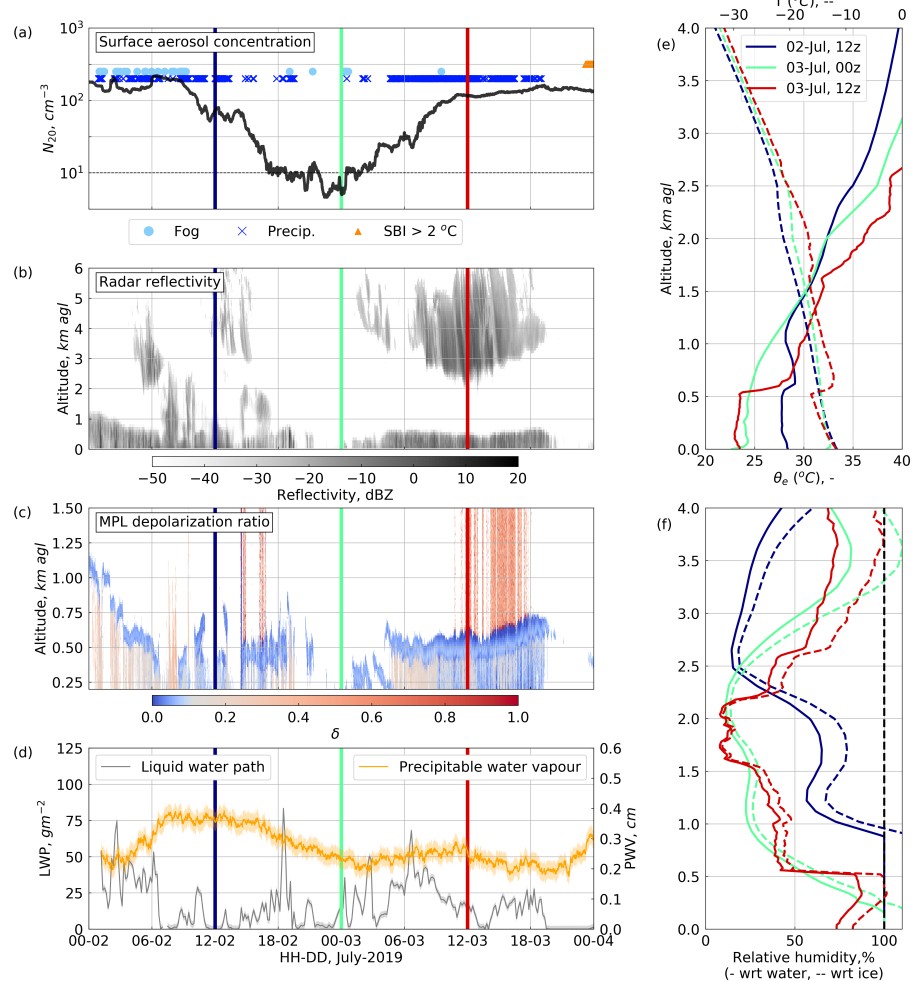

**Figure 11.** Conditions during the 03 July 2019 low aerosol case study. (a) Surface $N_{20}$ concentration (CPC), with occurrences of fog (OPC-N3), precipitation (POSS), and SBI > 3 °C events indicated. (b) Radar reflectivity (MMCR). (c) Lidar depolarisation ratio (MPL), blue colours represent liquid droplets and reds are ice crystals. (d) Column integrated liquid water path and precipitable water vapour (MWR). (e) Temperature (dashed) and equivalent potential temperature (solid) radiosonde profiles. (f) Relative humidity with respect to water (solid) and ice (dashed) from radiosonde profiles. The coloured vertical lines on the left-hand plots correspond to the time of each vertical radiosonde profile in the right-hand plots.

370    ## 4    Discussion

### 4.1    The seasonal cycle of surface aerosol concentrations at Summit

In 2019, the lowest monthly mean surface $N_{20}$ concentrations at Summit occurred in late February and early March, followed by a sharp increase throughout March and April, with the highest concentrations during the summer and early fall (Fig. 4).



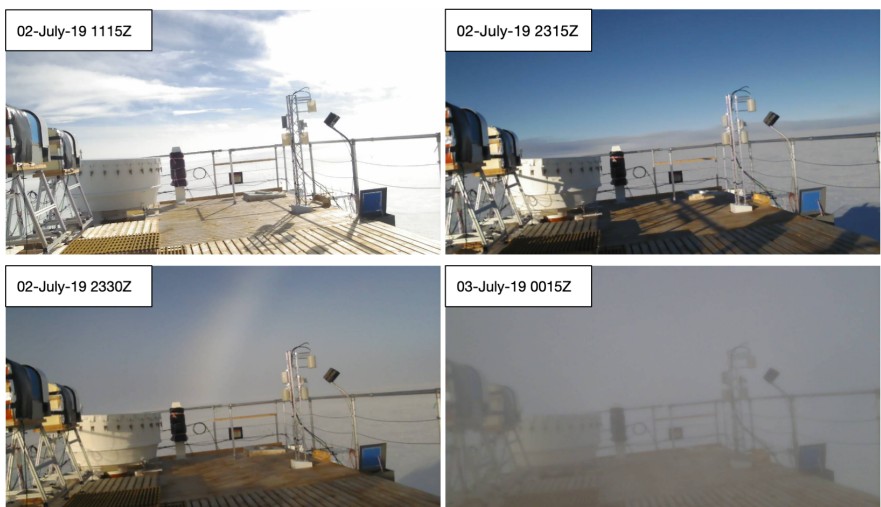

**Figure 12.** Photographs from a webcam oriented towards the eastern horizon on 02-03 July 2019. The aerosol inlet is visible mounted on the mast in the SE corner.

This seasonal cycle is consistent with multi year seasonal cycles of mineral particles in snowpit samples at Summit (Drab et al., 2002), and with measurements of bulk aerosol light scattering and absorption coefficients that are related to aerosol concentrations (Schmeisser et al., 2018). However, the seasonal cycle of surface aerosol concentrations at Summit is offset from the typical cycle of 'Arctic haze' (e.g. Shaw, 1995). Arctic haze is a common feature across the Arctic where anthropogenic pollutants build up in the winter resulting in maximum aerosol concentrations in early spring, followed by a sharp reduction of aerosols in the summer associated with an increase in wet deposition. The chemical and optical properties of Arctic haze are well characterised (e.g. Quinn et al., 2002), and there is no evidence that the Arctic haze reaches the top of the GrIS (Dibb, 2007; Schmeisser et al., 2018). Hirdman et al. (2009) used FLEXPART back trajectory simulations to show that surface aerosol concentrations at Summit are an order of magnitude less sensitive to surface emissions from within the Arctic compared to lower altitude Arctic sites. In contrast, the GrIS is more sensitive to aerosol sources above the boundary layer, often originating further south and descending to the GrIS via subsidence driven by radiative cooling (Stohl, 2006; Hirdman et al., 2009). In addition, the descent of aerosols to the surface from the upper troposphere, and the fact that precipitation in the accumulation zone of the GrIS occurs as ice phase year-round (e.g. Shupe et al., 2013b; Lenaerts et al., 2020; McIlhattan et al., 2020) and is less efficient at scavenging aerosol than liquid phase precipitation (Henning et al., 2004), suggests that aerosol losses to wet deposition during transport is reduced compared to sea level Arctic sites. Wet deposition is the main driver of the extremely low summer aerosol concentrations at sea level in the Arctic (Garrett et al., 2010; Browse et al., 2012), so the reduced wet deposition over the GrIS could explain the relatively high summertime aerosol concentrations.



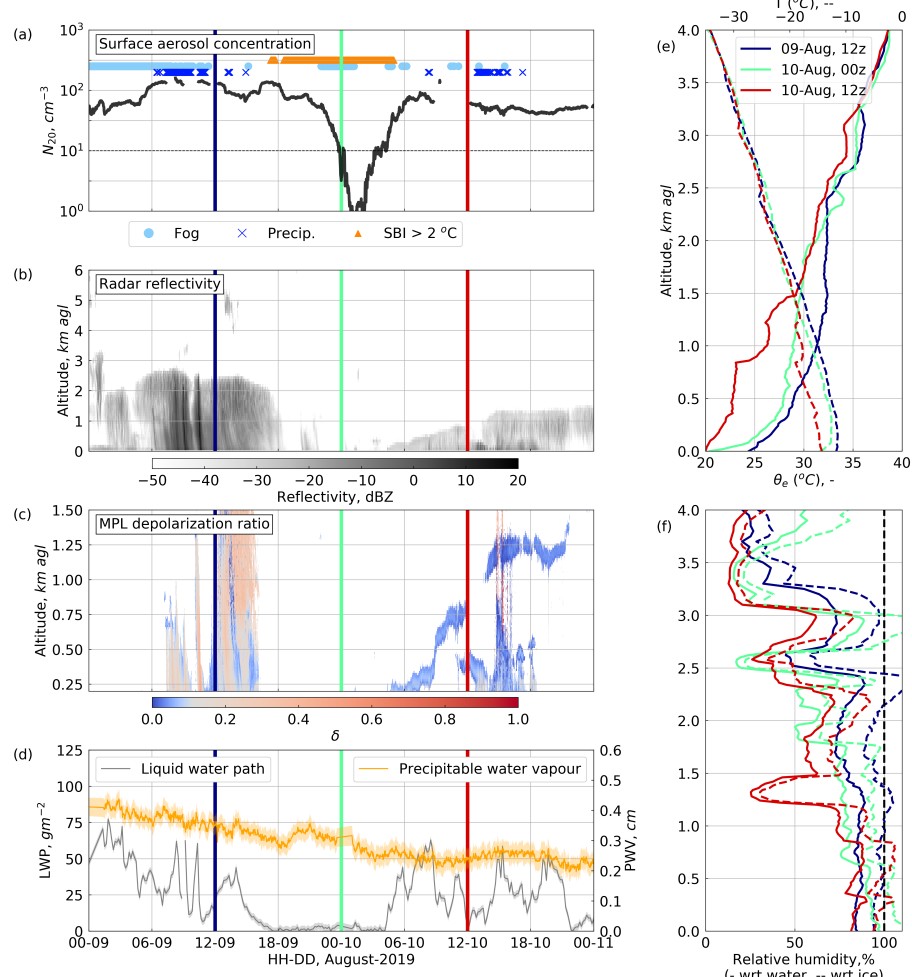

**Figure 13.** Same as Fig. 11 but during the 10 August 2019 low aerosol case study.

## 4.2 Controls on surface aerosol concentrations at Summit

The processes controlling surface aerosol concentrations over the central GrIS form a complex system, integrating local meteorological conditions, air mass history during aerosol transport, source regions and transport pathways. Fig. 16 illustrates some of the key components of this system, distinguishing between those processes that are supported by evidence in this study, and those for which uncertainties still remain.

Out of the four surface processes considered in this study (fog, SBIs, precipitation and BLSN), only fog events have a strong and consistent effect on measured surface $N_{20}$ concentrations. During the first three hours of a fog event, $N_{20}$ concentration was reduced by up to 30% in over half of the cases considered, with a median reduction of $\sim 20\%$. The effect of fog on surface aerosol concentrations is consistent with previous studies that were limited to the summer months (Bergin et al., 1994, 1995).





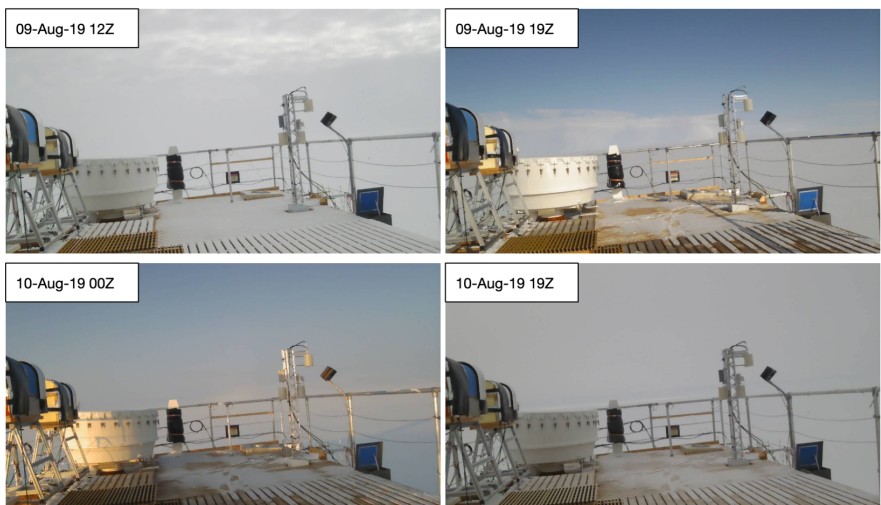

**Figure 14.** Photographs from a webcam oriented towards the eastern horizon on 09-10 August 2019. The aerosol inlet is visible mounted on the mast in the SE corner.

Because fog significantly modifies local surface aerosol concentrations, future studies should ensure that fog is accounted for before generalising sampled aerosol concentrations over wider regions or altitude ranges. Importantly, the observed decrease in aerosol number concentration during fog events is equal to the number concentration of activated CCN that grow too large to pass through the CPC inlet, and not necessarily the number concentration of aerosols deposited at the surface.

Despite the potential for SBIs to act as a barrier for turbulent mixing and hence reduce the rate that aerosols are transported down to the surface (Dibb et al., 1992; Li et al., 2019; Thomas et al., 2019), after the removal of events that were also effected by fog, we found no consistent change in surface $N_{20}$ concentration during the first 3 hours of SBIs > 0.23 $°C\,m^{-1}$ (Fig. 6b). There is no relationship between the change in $N_{20}$ concentration during the first 3 hours of the SBI event and the mean intensity of the SBI, which ranges between 0.23 $°C\,m^{-1}$ and 0.92 $°C\,m^{-1}$ (not shown). SBIs may have a more important role on surface aerosol concentrations over longer timescales, especially because the loss of aerosols to the surface by dry deposition is slow (Garrett et al., 2010). But because fog regularly forms during SBI events, it is difficult to isolate the influence of the SBI from the influence of fog scavenging on aerosol concentrations during longer events. SBIs may also contribute to observed reduction in surface $N_{20}$ concentration during fog events by restricting the turbulent mixing of high aerosol concentrations down to the surface. This study does not consider changes in mechanically induced turbulence over time or elevated temperature inversions, both of which could also effect the rate of turbulent mixing down to the surface. Further studies are required to understand the role of changes in turbulent mixing on controlling surface aerosol concentrations.

Surface $N_{20}$ concentrations also do not respond consistently to the precipitation events considered in this study (Fig. 6c). This is in agreement with Bergin et al. (1995), who did not observe a significant effect of precipitation on surface aerosol

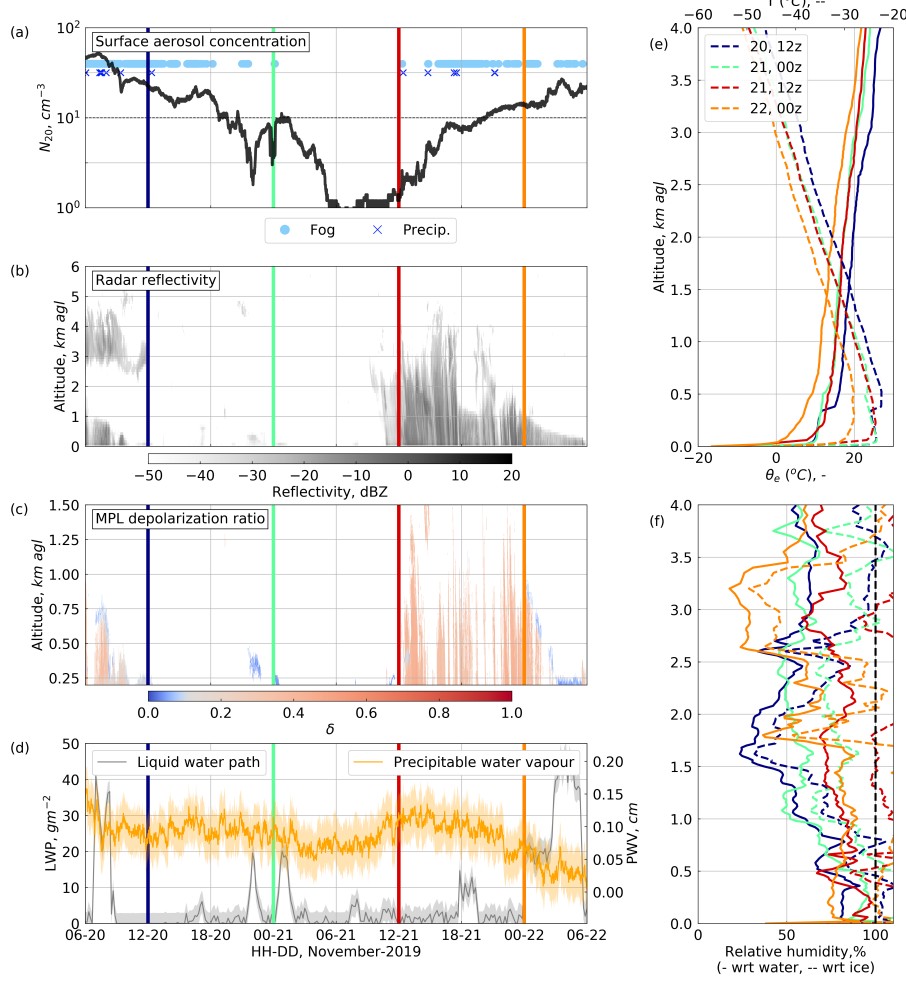

**Figure 15.** Same as Fig. 11 but during the 21 November 2019 low aerosol case study. Note that there are no SBI 'events' recorded during this period due to missing data, but that the radiosonde profiles indicate a constant shallow surface-based temperature inversion throughout.

concentrations at Summit during the summer. below-cloud scavenging rates are sensitive to a wide range of parameters that we do not consider here, including snow crystal size and habit, degree of riming, relative humidity and Reynolds number (Feng,

2009; Browse et al., 2012). We also do not distinguish between below-cloud precipitation and clear sky precipitation (diamond dust). However, although the rate of wet deposition might vary between events, below-cloud scavenging should reduce surface $N_{20}$ concentrations, and the fact that we do not consistently observe this suggests that other processes are acting to maintain surface aerosol concentrations during precipitation. For example, in both the 03 July 2019 and 21 November 2019 case studies (Section 3.4), aerosol concentrations increase during precipitation. One explanation for this could be due to the release of

aerosols near the surface via below-cloud evaporation of hydrometeors. Low-level mixed-phase clouds in particular can act to facilitate the transport of aerosols from the free troposphere into the boundary layer through entrainment and activation at





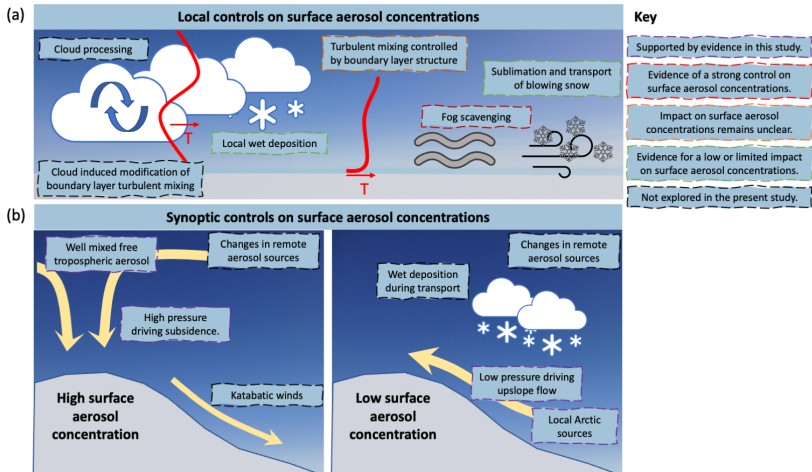

**Figure 16.** Conceptual model illustrating the key components controlling surface aerosol concentrations over the central GrIS, highlighting factors that are supported by evidence in this study and important areas for future research (see legend inset).

cloud top and release through evaporation at cloud base (Igel et al., 2017). At Summit, the majority of precipitation occurs in the presence of these low-level mixed-phase clouds (Pettersen et al., 2018). Clouds can also act to increase the efficiency of turbulent mixing down to the surface because below-cloud turbulent mixing driven by the sinking of radiatively cooled air near

the cloud top can extend down into the mechanically driven surface mixed layer (Brooks et al., 2017; Shupe et al., 2013a). Precipitation itself might also contribute to this increase in turbulent mixing via drag. Given that most of the aerosols arriving at Summit descend from the free troposphere (Hirdman et al., 2009), the role of clouds in the transport of aerosols into the boundary layer of the GrIS is an important area for future research.

     BLSN may have competing influences on surface aerosol concentrations, because BLSN can scavenge aerosol via impaction

and interception but can also re-release aerosols through sublimation (e.g. Frey et al., 2020). In this study we find no evidence of a consistent net effect of BLSN on $N_{20}$ concentration during the first 3 hours of a BLSN event. Note that it is not possible to determine whether or not fog is also present during blowing snow events.

     Synoptic conditions play an important role in controlling surface $N_{20}$ concentrations at Summit, with most anomalously high aerosol concentration events coinciding with anomalously high surface pressure during 2019 (Fig. 7). The difference in aerosol

emission sensitivity and transport pathway simulations between anomalously high and low aerosol concentration events (Fig. 9), combined with the difference in mean synoptic anomalies (Fig. 8), imply that high surface aerosol concentrations occur at Summit when air is transported down to the surface from high in the free troposphere, associated with subsidence related to anomalous strong high pressure systems over Greenland (Fig. 16b). This free tropospheric air is sensitive to aerosol emissions from mid and low latitudes that can release particles high into the atmosphere where they remain suspended for long periods

of time (i.e. > 20 days) (Stohl, 2006; Hirdman et al., 2009; Roiger et al., 2011). This result is consistent with previous studies investigating the transport pathways of aerosols that arrive at Summit. Both Hirdman et al. (2009) and Schmeisser et al. (2018)



conclude that because on average the majority of air arriving at Summit has only been in contact with the surface over the ice sheet itself, aerosols measured at Summit must have descended from the free troposphere after transportation at high altitudes over time scales > 20 days. Persistent anomalously high geopotential heights over central Greenland are also associated with

the occurrence of precipitating low-level mixed-phase stratocumulous clouds (McIlhattan et al., 2020) that can encourage the transport of aerosols from the free tropospheric into the boundary layer (Igel et al., 2017).

Anomalously low aerosol concentrations typically occur in the presence of anomalous cyclonic circulation and low geopotential heights off the south east coast of Greenland (Fig. 8b) that drive air up to the top of the GrIS from the coast and surrounding ocean (Fig. 9b, 16b). During such events, adiabatic cooling due to orographic lifting as the air is advected up the

GrIS results in increased condensation and precipitation (Schuenemann et al., 2009). These conditions are associated with deep glaciated clouds advecting over Summit from south east Greenland (Pettersen et al., 2018). The associated increase in wet deposition en-route to Summit could contribute to the relatively low aerosol concentrations. These events are more common in the winter season, when the north Atlantic storm track is more active (Schuenemann et al., 2009). Hogan et al. (1984) also reached a similar conclusion based on surface aerosol measurements at DYE III during the summer; they observed that low aerosol

concentrations followed moist upslope flow and precipitation driven by a low pressure system to the south of Greenland, and that concentrations increased after the establishment of a high pressure system and downslope flow.

This study does not consider changes in emission or removal rates along the aerosol transport pathway. Emission rates vary seasonally within the Arctic due to changes in ice cover and biomass burning (Willis et al., 2018), and isolated events such as volcanic eruptions can have large impacts on background aerosol concentrations (e.g. Friberg et al., 2015). Removal

rates vary along a particular transport pathway with changes in precipitation amount and phase (Garrett et al., 2010; Browse et al., 2012). Given this, it is quite remarkable that the relationship between anomalous aerosol concentrations at Summit and anomalous synoptic conditions is so evident. The strength of this relationship implies that future changes in Arctic large scale circulation could effect aerosol concentrations and aerosol-cloud-radiation interactions over the GrIS. In particular, changes in the frequency of storms moving up the southeast coast of Greenland (Ulbrich et al., 2008) or the position of the Icelandic low

(Berdahl et al., 2018) might effect the frequency of extremely low aerosol concentration events over the central GrIS.

At sea level Arctic sites (both marine and coastal), the extremely low aerosol concentrations observed in the summer are largely attributed to increases in wet deposition (Garrett et al., 2010; Browse et al., 2012). An important distinction between Summit (where the 0 °C isotherm is always below the surface except in extreme situations; Shupe et al., 2013b; Bennartz et al., 2013) and sea level Arctic sites, is that Summit does not experience rain during the summer. The fact that mean monthly

surface $N_{20}$ concentrations are relatively high in the summer at Summit could be related to the fact that wet deposition is much less efficient in ice-bearing clouds (Henning et al., 2004). In this case, future increases in the height of the 0 °C isotherm over the GrIS could result in lower summertime aerosol concentrations.

### 4.3 Potential for cloud formation to be limited by low CCN concentrations and discussion of case studies

Surface $N_{20}$ concentrations fell below 10 cm$^{-3}$ on multiple occasions year-round at Summit in 2019 (Fig. 4). Because CCN

are a subset of total condensation nuclei concentration, it is likely that CCN concentrations fall low enough to limit cloud and





fog formation, based on approximate threshold estimates determined from past observational and modelling studies over the Arctic Ocean (Mauritsen et al., 2011; Stevens et al., 2018). The ratio of $CCN/N_{10}$ at a supersaturation of 0.55% measured over Summit during a research flight in 2008 was 0.52 (Lathem et al., 2013), which is similar to the mean $CCN/N_{10}$ ratio observed at the Zeppelin Observatory in Svalbard outside of the Arctic haze season (Jung et al., 2018). However this ratio is a function of

supersaturation, and at very high supersaturations (that can occur under extremely low CCN concentrations) small particles that do not typically act as CCN can activate (Leaitch et al., 2016; Baccarini et al., 2020). If we make the assumption that all CCN are activated after the first 3 hours of fog formation during the events in Fig. 6a, the fact that we see a median 20% reduction in total $N_{20}$ concentrations during these events implies a $CCN/N_{20}$ ratio of 0.2, and for the individual event example in Fig. 3, the $CCN/N_{20}$ would have been 0.46. Using the more conservative ratio estimation of 0.46, surface CCN concentrations will

have fallen below 10 cm$^{-3}$ for 46 days or 15% of the measurement period during 2019. Because supercooled liquid fog can have a large effect on surface radiative fluxes at Summit (liquid fog at Summit has an average total (SW+LW) cloud radiative forcing of 26.1 W m$^{-2}$ compared to clear skies, Cox et al., 2019), if fog formation is limited by low CCN concentrations, this could have an important effect on the ice sheet surface energy budget, especially over individual events which can play a role in pre-conditioning the snow surface in advance of melt (Miller et al., 2017). The same could be true for clouds where surface

concentrations are representative of CCN concentrations at cloud level. For example, the exceptional July 2012 Greenland melt event was enhanced by the presence of low-level mixed-phase clouds with a LWP of $\sim$ 30 g m$^{-2}$ (Bennartz et al., 2013), in this case, if small changes in CCN concentrations acted to either increase or decrease the cloud LWP, they could have controlled the presence versus absence of surface melt.

In our first case study of a potential CCN-limited event (3 July 2019; Fig. 11), the potential temperature profile indicates

that the boundary layer is initially well mixed up to cloud height (Fig. 11e) suggesting that surface aerosol concentrations were likely representative of concentrations in the cloud (Creamean et al., 2021). By the time the cloud had cleared and surface $N_{20}$ concentrations fell to $<$ 10 cm$^{-3}$ this coupling had broken down and a region of high static stability had formed near the surface (Fig. 11e), likely due to increased radiative cooling at the surface as the cloud thinned. Although we cannot tell when the decoupling between the cloud layer and the surface occurred, the fact that there was no cloud and unlimited visibility

despite the lowest 200 m above the surface remaining saturated with respect to water strongly implies fog droplet formation, and possibly cloud formation too, was limited by low CCN concentrations in this case. Fog was detected at the surface again as soon as aerosol concentrations began to increase, shortly followed by the return of a shallow-mixed-phase cloud and coupled boundary layer.

The second case study (10 August 2019) was different in that the initial mixed-phase cloud layer was much deeper (2.5

km compared to $\sim$ 0.75 km) and the cloud layer remained decoupled from the surface throughout (Fig. 13e). Surface $N_{20}$ concentrations remained relatively stable as the cloud thinned and did not begin to decrease until the establishment of a strong surface temperature inversion (Fig. 13a). The surface temperature inversion itself was likely strengthened by decreased long-wave radiative warming at the surface after the dissipation of the cloud. This surface cooling would have increased relative humidity, resulting in the surface fog detected at 22Z, which accelerated the decrease in aerosol concentrations. However, the

fog dissipates as soon as surface $N_{20}$ concentrations fall below 10 cm$^{-3}$ despite the surface RH remaining close to supersat-



uration (Fig. 13f), so again fog formation was likely limited by low aerosol concentrations in this case. As soon as aerosol concentrations increase above 10 cm$^{-3}$ again, there is a sharp increase in liquid water path (Fig. 13d) and fog is detected at the surface again.

The third case study (21 November 2019) occurs close to the annual solar minimum. Surface temperatures were much colder than the summer cases and although data from the 15 m temperature sensor were unavailable, the radiosonde profiles show that SBIs are persistent throughout (Fig. 15e). The OPC-N3 detected fog both before and after surface $N_{20}$ concentrations fell below 10 cm$^{-3}$, but again, whilst surface $N_{20}$ concentrations were $< 10$ cm$^{-3}$, liquid water path remained close to zero, and no fog was detected (Fig. 15). The cloud that passed over Summit at 21Z on 21 November 2019, when surface $N_{20}$ concentrations were at their minimum, consisted solely of ice. Surface $N_{20}$ concentrations begin to increase as soon as the cloud passed over

the station, this could have been related to increased turbulent mixing driven by overturning or precipitation within the cloud, or by a reduction in the strength of the SBI due to a decrease in longwave cooling of the surface (e.g., Shupe et al., 2013a). Coinciding with surface $N_{20}$ concentrations increasing above 10 cm$^{-3}$, there was a sudden increase in liquid water path and a decrease in cloud top height from $\sim 1.5$ km to 0.5 km (Fig. 15). In this case, it is unclear whether the increase in $N_{20}$ (and hence CCN) concentration drove the formation of liquid droplets within the cloud, or whether the cloud was responsible for

increasing the mixing of high $N_{20}$ concentrations down to the surface.

For all three cases, decreasing aerosol concentrations were associated with a reduction in cloud cover, and the reverse was also true. However, differences in timing and boundary layer structure imply that different processes were involved in each case. This demonstrates that it is not sufficient to use simple correlations between cloud properties and aerosol concentrations to investigate cloud-aerosol interactions, since there are many additional confounding variables. Although we cannot delineate

the individual drivers of the changes in surface aerosol concentrations during these case studies based purely on observations, the near-zero liquid water path is convincing evidence that low CCN concentrations are limiting the formation of liquid water droplets at the surface despite supersaturation when surface $N_{20}$ concentrations are $< 10$ cm$^{-3}$ in all three case studies (i.e. fog formation). One of the biggest challenges in assessing the role of aerosols in controlling cloud properties is that it is not possible to disentangle what is driving changes versus what is responding from observations alone. Process-based model

studies constrained by observations are required to address this issue.

Finally, for all three case studies, back trajectory simulations indicate that aerosols were transported upslope to Summit from lower elevations (Fig. 10), and two of the cases (July and November) occurred in the presence of cyclonic circulation off the southeast coast of Greenland - the typical synoptic condition associated with anomalously low aerosol concentrations at Summit (Fig. 8). Although the simulated aerosol source regions are all from high latitudes ($> 50°$N; Fig. 10), they originate

from very different directions in each case (Siberia on 3 July 2019, the Canadian Archipelago on 10 August 2019, and south west of Greenland on 21 November 2019). This suggests that the upslope transport pathway to Summit, which is strongly linked to precipitation over the GrIS (Schuenemann et al., 2009) and notably from glaciated as opposed to mixed-phase clouds (Pettersen et al., 2018), has a stronger influence on surface $N_{20}$ concentrations than the source region. These upslope flow enhanced precipitation events are also coupled to anomalously warm temperatures over the GrIS, which likely results in a

higher percentage of rain (and hence increased wet deposition) en-route to Summit (Pettersen et al., in review). These results





imply that increased wet deposition during transport may play a large role in driving CCN concentrations below the threshold where they can sustain cloud formation. The role of wet deposition in controlling aerosol concentrations over the central GrIS is therefore an important area for future research.

## 5   Summary and conclusions

This study presents the first full year of surface aerosol number concentration measurements from the central Greenland Ice Sheet and assesses the local and synoptic controls on surface $N_{20}$ concentration. In 2019, the minimum aerosol concentrations occur in February (which has a monthly average concentration of just 18 cm$^{-3}$, and a standard deviation, $\sigma_N = 16$ cm$^{-3}$), and the maximum concentrations occur in April and May (monthly average concentrations of 247 cm$^{-3}$, $\sigma_N = 130$ cm$^{-3}$ and 206 cm$^{-3}$, $\sigma_N = 165$ cm$^{-3}$, respectively). Between May and October, concentrations remain on the order of 100 cm$^{-3}$ before

they decrease again between October and December. This seasonal cycle is opposite to that of sea level Arctic sites which experience minimum surface aerosol concentrations in the summer (Schmeisser et al., 2018), implying that the processes controlling aerosol concentration over central Greenland are distinct from other parts of the Arctic.

Changes in synoptic conditions strongly control surface $N_{20}$ concentrations, with almost all anomalously high $N_{20}$ concentration events associated with anomalously high surface pressure over Summit. High surface $N_{20}$ concentrations occur under

anomalously high geopotential heights and strong anticyclonic circulation over Greenland, which act to enhance the descent of free tropospheric air to the ice sheet surface. Low surface $N_{20}$ concentrations occur in the presence of anomalous cyclonic circulation over south east Greenland, when low pressure systems drive up slope flow that is associated with increased precipitation (Schuenemann et al., 2009; Pettersen et al., 2018). Below average aerosol concentrations occur more often in the winter, when the frequency of low pressure systems driven by the North Atlantic storm track increases (Schuenemann et al.,

2009). The distinction between upslope flow and descent from higher altitudes appears to be a stronger control on surface $N_{20}$ concentrations than aerosol source region, suggesting an important role for wet deposition along aerosol transport pathways.

We find that fog strongly effects surface aerosol concentration measurements, in agreement with previous studies that look at isolated events during the summer (Bergin et al., 1994, 1995). On average, there is a 20% reduction in surface $N_{20}$ concentrations after the first 3 h of a fog event. Because fog significantly modifies local surface aerosol concentrations, future

studies should ensure that fog is accounted for before generalising sampled aerosol concentrations over wider regions or altitude ranges. In contrast, precipitation, blowing snow, and strong surface-based temperature inversions ($> 0.23$ °C m$^{-1}$) do not have a consistent effect on surface $N_{20}$ concentrations during the first 3 h of the event. Competing influences of advection, or either cloud or mechanically induced changes in the turbulent structure of the boundary layer, might play roles in modulating aerosol concentrations during these events and are not considered in this study.

This study uses a conservative estimate to determine that surface aerosol concentrations low enough to limit cloud and or fog formation (based on observations and model simulations over the Arctic ocean; Mauritsen et al., 2011; Stevens et al., 2018) do occur in both winter and summer over the central GrIS. However, long term vertical profiles of CCN concentrations are necessary to determine how often this is relevant at cloud height. Although practically difficult, continuous vertical profiles





of aerosol concentrations above the GrIS are essential for understanding the interaction between clouds, aerosols, and the ice
sheet surface energy budget, and should be a priority for future campaigns. Vertical aerosol profiles are particularly important
over the central GrIS where most of the aerosol arriving at the surface descends from higher elevations in the free troposphere
(Hirdman et al., 2009; Schmeisser et al., 2018, this study). The unique transport pathway and resulting seasonal cycle of
aerosols over the central GrIS demonstrate that observations of aerosol properties at sea level Arctic sites cannot be generalised
over the GrIS, in agreement with previous studies (e.g. Hirdman et al., 2009; Schmeisser et al., 2018; Schmale et al., 2021).

*Data availability.* All data are publicly available. ICECAPS-ACE aerosol measurements and multi level temperature sensor data can be
accessed through the CEDA archive at http://catalogue.ceda.ac.uk/uuid/f06c6aa727404ca788ee3dd0515ea61a. NOAA GML meteorologi-
cal data are available at ftp://aftp.cmdl.noaa.gov/data/645met/sum/ (last access: March 2021). All additional ICECAPS data are available
from the Arctic Data Center: MMCR (doi:10.18739/A2Q52FD4V), MPL (doi:10.18739/A2862BC30), POSS (doi:10.18739/A2GQ6R30G),
HATPRO and MWRHF (doi:10.18739/A2TX3568P) and radiosonde profiles (doi:10.18739/A20P0WR53). ERA5 reanalysis data are made
available by the European Centre for Medium-Range Weather Forecasts (ECMWF) and can be accessed at the Copernicus Climate Data
Store, DOI: 10.24381/cds.bd0915c6.

*Author contributions.* The original ICECAPS project proposal was conceived by MDS, VPW, DDT and RB. HG led data collection and
curation of ICECAPS-ACE aerosol data supervised by RRN and IMB. HG led the analysis with contributions from RRN, IMB, BJM, CP,
MDS and CJC. HG prepared the manuscript with contributions from all co-authors.

*Competing interests.* The authors declare that they have no conflict of interest.

*Acknowledgements.* The efforts of technicians at Summit Station and science support provided by Polar Field Services were crucial to main-
taining data quality and continuity at Summit. We acknowledge Bethany Wyld for assistance during the ICECAPS-ACE field installation,
and Richard Rigby for his support setting up and running the FLEXPART model. We also gratefully acknowledge support from the National
Centre of Atmospheric Science (NCAS), the NCAS Atmospheric Measurement and Observation Facility, and notably Dr. Barbara Brooks
for support in instrument troubleshooting and data quality control. ICECAPS is a long-term research program with a large number of collab-
orators and we are grateful for all their efforts in developing and maintaining the various instruments and data products used in this study.
Financial support for ICECAPS-ACE was provided by NSFGEO-NERC grant 1801477, and HG was funded by the NERC SPHERES DTP
grant number NE/L002574/1. MDS and WDN were supported by the National Science Foundation grant OPP1801477.





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





**Table 1.** Measurements used in this study, references provide additional instrument information and methodologies for derived parameters.

| Instrument | Measured / *derived* parameters (used in this study) | Data availability (used in this study) | Reference |
|---|---|---|---|
| Condensation Particle Counter (CPC) | Ambient condensation nuclei number concentration > 5 nm | Feb '19-May '20 (exc. Jan '20) | Guy et al. (2020) |
| Alphasense Optical Particle Counter (OPC-N3) | Aerosol size distribution 0.35 to 40 $\mu m$ | Jun-Dec '19 | Crilley et al. (2018) |
| NOAA Met. suite | 10 m wind speed and direction, Surface pressure | Feb '19-May 20' | GMLMET (2021) |
| Vaisala HMP155 T/RH probe | 2 m air temperature, 15 m air temperature | Jun-Oct '19 | Guy et al. (2020) |
| Precipitation Occurrence Sensor System (POSS), X-band (10.5 GHz) | *Precipitation occurrence (POSS power unit)* | Mar-Dec '19 . | Sheppard and Joe (2008) |
| Millimeter cloud radar (MMCR) Ka band (35 GHz) | Radar reflectivity | Case studies only | Moran et al. (1998) |
| Radiosondes (00Z and 12Z) | Vertical temperature and humidity profiles | Case studies only | Shupe et al. (2013b) |
| HATPro and MWRHF microwave radiometers (23, 21, 90 and 150 GHz) | *Liquid water path,* *precipitable water vapour* | Case studies only | Turner et al. (2007) Shupe et al. (2013b) |
| Micropulse lidar (MPL) | *Lidar depolarisation ratio* | Case studies only | Flynna et al. (2007) |