# Peer review of "Controls on surface aerosol number concentrations and aerosol-limited cloud regimes over the central Greenland Ice Sheet."

_Atmospheric Chemistry and Physics, 2021_

## Author Response (AR1)

**Acp-2021-491: Response to reviewers**

09 September 2021

**The co-authors would like to thank both reviewers for their feedback and thoughtful suggestions. We have responded to each comment below.**

Original reviewer comments are included in *blue italics*. Co-author responses are in simple text, and quotations from the revised manuscript are in *black italics*.

**Reviewer I**

*This is a very well written manuscript that presents a year's worth of surface aerosol concentrations at Summit Station Greenland along with in-depth analysis of the processes that control the surface aerosol concentration. I have only a few minor comments:*

1. *My biggest comment is simply that this paper is long. I felt that the Discussion section was largely repetitive (though certainly not entirely) to discussion that had already occurred in the Results section.*

We have done our best to shorten the paper and reduce repetition by removing any speculative discussion from the results section and reducing the length of the discussion section. Notably, we removed the detailed discussion of each case study from section 4 and focused mainly on the similarities and differences between the three cases. Overall this has reduced the length of the manuscript by 825 words, with the key points and conclusions unchanged. We believe this has improved the readability and clarified the overarching messages in this paper.

2. *Lines 8 and 290: Be sure to state "anomalous" cyclonic circulation.*

We have specified 'anomalous' at these two points.

3. *Lines 59-62: The implication here is that the WBF process will be active whenever ice crystals and cloud droplets are co-located. This is not the case since air can be either supersaturated or subsaturated with respect to both liquid and ice simultaneously. Please just modify the sentence to avoid the implication.*

We have modified this sentence to the following:

**Lines 58 to 62:** *"...once ice crystals are present in a supercooled cloud, the lower saturation vapor pressure of ice versus liquid water results in the preferential growth of ice crystals at the expense of liquid droplets when the environment is subsaturated with respect to water but supersaturated with respect to ice. This is known as the Wegener-Bergeron-Findeisen (WBF) process, the result of which is a decrease in LWP as droplets evaporate and an increase in precipitation due to the*

*growth of relatively large ice crystals, ultimately leading to cloud dissipation (e.g. Lohmann and Feichter, 2005)."*

> 4. *Lines 265-266 and 436-437: The authors note that it is not possible to determine if blowing snow and fog (presumably supercooled liquid fog since that is the kind of fog that the authors have discussed) occur simultaneously. That may be true based on the instrumentation. But if fog were present at the start of a blowing snow event, wouldn't we expect the blowing snow to rime the supercooled liquid or potentially eliminate it through the WBF process? It seems unlikely to me that we would have both fog and blowing snow for very long.*

This is a good point and should be noted. We deleted the sentence on line 436, as it was a repetition, and have added the following sentence:

**Lines 256-257.:** *"However, because of the high concentrations of ice crystals during blowing snow events, any supercooled water droplets are likely to be removed either through riming or the WBF process."*

> 5. *The authors present three case studies of low N20 and show that in all three cases there is "near-zero" liquid water when N20 is < 10 cm-3. Is there ever non-near-zero liquid when N20 < 10 cm-3? Or maybe there simply aren't enough events for the answer to this question to be meaningful.*

We look specifically for events where exceptionally low surface aerosol particle concentrations coincided with cloud dissipation as 'possible candidates' for low aerosol particle concentrations limiting cloud formation. We know that a lot of the time, surface aerosol particle concentrations are unlikely to be representative of concentrations at cloud level, especially when persistent strong surface temperature inversions decouple the surface layer from the cloud layer. On such occasions, we might expect events with exceptionally low particle concentrations at the surface below cloud layers containing liquid water.

In total (from Fig. 4), there are 2,533 1-minute mean data points where N20 falls below 10 cm-3, and a LWP retrieval from the MWR is available (Feb to Nov 2019 only). Aside from the case studies discussed in the publication, 91% of these cases have a LWP < 10 g m-2. The other 9% occur across four events in February and March 2019. During each of these events, a cloud layer was present with a base height between 250 and 1000 m above the surface which was saturated with respect to water and up to ~20 C warmer than the surface. Therefore, in these cases, it is not possible to say whether the measured aerosol particle concentrations were representative of concentrations in the cloud.

To help clarify this for other readers, we have added the following sentences to the text:

**Lines 492 to 495:** *"Note the only other events where we observed N20 < 10 cm−3 and LWP > 10 g m−2 occurred in February and March 2019 and were associated with clouds with base heights between 250 and 1,000 m above the surface. The static stability of the surface layer in these cases means it is not possible to know whether the surface N20 was representative of aerosol particle concentrations in the cloud layer."*

Reviewer II

*The manuscript presented by Guy and coauthors is of both excellent scientific quality and presented in an excellent language. It features highly valuable measurements, that fill a "blank spot on the map" in the understanding of the aerosol-cloud-climate system in the Arctic. The methodology is of sound quality. It was both interesting and a pleasure to read.*

*General comment:*

*The manuscript is somewhat long in its current form, and the central points and conclusions would be communicated better, if the manuscript was shortened, where possible. The case studies are presented with a lot of information and very dense plots, and the discussion section feels unnecessarily lengthy. I suggest that the authors consider where the manuscript can be shortened if/where they find it suitable.*

We have done our best to shorten the paper and reduce repetition by removing any speculative discussion from the results section and reducing the length of the discussion section. Notably, we removed the detailed discussion of each case study from section 4 and focused only on the similarities and differences between the three cases. Overall this has reduced the length of the manuscript by 825 words, with the key points and conclusions unchanged. We believe this has improved the readability and clarified the overarching messages in this paper. This is also in line with the changes made to address the similar concerns of Reviewer I.

*Specific comments:*

*Lines 5-6, 560: The statement that the annual pattern of N20 is opposite that of other Arctic sites, is speculative at best. Please see Freud et al. 2017, Atmos. Chem. Phys., 17, 8101–8128, 2017 in this matter. There is also not a 1-to-1 comparison between N20 used here, and the lower cut-offs and size distributions reported by different authors.*

Yes, I see that this is poorly phrased. We have deleted this sentence from the abstract and replaced the sentence at line 560 with the following:

**Lines 515 to 516:** "*This seasonal cycle is distinct from many sea level Arctic sites that observe minimum surface aerosol concentrations in the summer (Freud et al., 2017; Schmeisser et al., 2018)*"

*Lines 62-63: There is only a very weak link between CCN and INP, as ice nucleation and the initiation of freezing can happen through several mechanisms, that differ*

*from the condensation of a cloud droplet. The sentence can be read, is if INP is a subset of CCN. This is no the case.*

We have rephrased this sentence to the following :

**Lines 62-64.:** *" INP concentrations are typically orders of magnitude lower than CCN concentrations, and are particularly low in the Arctic based on limited existing measurements (~$10^{-7}$ to $10^{-5}$ cm$^{-3}$, Wex et al., 2019)."*

*Lines 146-149: There is no information presented about the particle sizes samples by the CPC. While this is perfectly legitimate, there are reports of elevated sub-30 nm particle concentrations af Summit (Ziemba et al. 2010, Atmospheric Environment 44 (2010) 1649-1657). Combined with the size dependent losses of the CPC sampling inlet, this will lead to underestimation of the reported N20 in some occasions. Can the authors elaborate on the uncertainties expected to be affiliated with the reported N20.*

We had not seen the Ziemba et al. (2010) paper, so are extremely grateful to you for bringing this to our attention. We have added this reference at several points throughout the paper: lines 105, 152, 156 and 396.

From Fig. 3 in Ziemba et al. (2010), we have estimated the expected N20 undercount for our CPC during that measurement period to be 27% (see the figure sf1.png in the attached supplements for an illustration of this). The fact that our CPC will count some particles between 5 and 20 nm might reduce the absolute value of this bias. Still, we expect this to be small given the relatively low observed concentrations < 20 nm coupled with the relatively low CPC collection efficiency in this size range. Of course, we would expect the undercount to vary a lot even within the short (1-month) sample period reported in the Ziemba et al. (2010) study; with a higher undercount when there are larger concentrations of Aitken and nucleation mode aerosols and a very minimal undercount when the size distribution is dominated by accumulation mode aerosol. Based on studies at other (admittedly at non-comparable Arctic sites) and the supposition of Ziemba et al. (2010) that the increased aged-nucleation mode aerosol events are related to photochemical processes, it seems reasonable to assume that there are large seasonal and sub-seasonal differences in this size distribution. Due to the completely unknown nature of the annual variability in the near-surface aerosol particle size distribution at Summit, it is not possible to robustly estimate an uncertainty in the N20 bias. To address this in the text, we have added the following paragraph in section 2.1:

**Lines 152 to 158:** *"Ziemba et al. (2010) took measurements of surface aerosol particle size distribution between 5.5 and 195 nm at Summit in May and June 2007. Their observations suggest that high concentrations of nucleation mode particles (< 30 nm diameter) occur periodically during the Summer at Summit. The reduced collection efficiency of our CPC between 20 and 40 nm would have resulted in an*

*under-count of the total N20 concentration by up to 27% during the 2007 measurement period reported by Ziemba et al. (2010), but only 8% in the accumulation mode (100 to 200 nm). The concentration of ultrafine particles (< 100 nm diameter) at Summit likely varies seasonally and on shorter timescales. In the absence of year-round measurements of particle size distribution at Summit, it is not possible to fully quantify the uncertainties in N20 reported here."*

[Figure]

*Line 176: "00Z, 12", and 18Z daily" Can it be expected that the majority of readers will be familiar with time given as zulu time?*

We have corrected these times to HH:MM UTC as per the ACP terminology guidelines. We have also applied this correction elsewhere in the manuscript; in Table 1, in the caption for Fig. 3 and within the plots of Fig. 3 and Figs 11, 13 and 15. In section 3.4, We added line 302, *"All times throughout the discussion of these case studies are given in UTC."* and removed the timezone reference in the text to facilitate ease of reading.

*Line 216-218: Is an assumed mean diameter of 2.5 μm justified?  I would expect 2.5 μm particles to only contribute minimally to N20. The authors state, that parameter variations have been made. Would it be more relevant to include FLEXPART calculations based on a particle size that is closer to the expected mean particle diameter?*

In checking this I found that reporting 2.5 μm was my mistake. The FLEXPART default 'aerosol tracer' species actually uses an assumed mean particle diameter of 0.25 μm, and this is the species that we used in our model runs. The full parameter selection for this species is listed here: https://www.flexpart.eu/browser/flexpart.git/options/SPECIES/SPECIES_025?rev=9f59af6cf93ec051bd908d238d54eb73289e8f23.

We have made this correction and justified this choice in the text as follows:

**Lines 222 to 227:** *"Due to limited prior information about aerosols at Summit, we used the default aerosol tracer species, which assumes a particle mean diameter of 0.25 μm, a density of 1400 kg m−3, and water and ice nucleation efficiencies of 0.9 and 0.1 respectively. Particles of 0.25 μm diameter are efficiently measured by the CPC at Summit (Fig. 2), fall within the typical size range of Arctic CCN (e.g. Jung et al., 2018), and have the relatively long atmosphere lifetimes necessary for advection over the GrIS ( > 10 days in the middle-upper troposphere; Jaenicke, 1990)."*

*Line 243: While eventually not relevant in the whole picture. The process described in Ziemba et al. 2010 could contribute to CCN relevant N20 locally.*

Thank you, we were not aware of these observations. We have added the following sentences in the discussion section:

**Lines 391 to 395:** *"We have made the assumption that there are no local sources of aerosol at the surface. There is a possibility that particle growth via condensation of precursor gases, possibly released from organic material in the snowpack, could occasionally contribute to near-surface CCN concentrations (Ziemba et al., 2010). We do not consider this process in the present study, but the contribution of ultra-fine particle growth to CCN concentrations over the GrIS remains unclear and warrants further investigation."*

*Figure 5: While obvious, formally the abbreviation Precip has not been introduced. The short form is not required here, as there is sufficient space.*

We have changed this to precipitation in Fig. 5 and Fig. 6, and in the figure captions.

*Figure 6: How can the pink line increase in some instances? Also, "Precip" is used again here.*

We have changed 'precip' to 'precipitation' here.

Originally, the small increases in the pink line arose from missing data points (NaNs in the CPC data that meant an event that was still ongoing might not be counted at one time step but then be counted again at a future timestep). However this does seem unnecessarily confusing for the reader. We have changed this plot so that the pink line is simply the number of events that reach a certain age - therefore it cannot increase, as per your interpretation. This makes the plot more intuitive, easier to interpret, and does not change the overall picture.

*Figre 7a: This is a  dense, and eventually unnecessarily complicated plot. Could a scatterplot of surface pressure anormaly vs. N20 anormaly be a simpler and clearer way of showing the correlation?*

There are a few reasons why we chose to show Fig. 7a as opposed to a surface pressure anomaly vs N20 anomaly scatter plot with correlation analysis.

Firstly, a simple linear correlation is actually not very meaningful with this dataset because the N20 anomalies are not symmetrical (i.e. N20 concentration has a lower limit of 0 but no upper limit, so the 'low' anomalies are restricted in magnitude in a way that the high anomalies, and the surface pressure anomalies, are not).

Secondly, it is plausible that there might be a phase shift in the relationship between surface pressure changes and aerosol changes, Fig. 7a allows the reader to inspect the timing of this relationship themselves in a way that would not be possible with a simple scatter plot.

Finally, an important purpose of Fig. 7a is to highlight the timing and duration of the anomalous 'high' and 'low' N20 concentration events that are then used to generate Fig. 8 and Fig. 9. We believe that this is important because it visually demonstrates that the 'high' and 'low' events are evenly distributed throughout the year and hence Fig. 8 and Fig. 9 are showing a signal that is independent of seasonal variability.

We have included the simple scatter plot in the attached supplement for your reference (sf2.png). We also added the following line to the text to emphasize the significance of showing the timing of the events in Fig. 7a.

**Line 273-275 :** *"The resulting high and low N20 events are highlighted in Fig. 7a, and are spread evenly throughout the annual cycle (15 high events and 14 low events)"*

[Figure]

*Line 296: It is implied that the particles that are referred to, as being primarily above the boundary layer, are simulated particles. This could be stated explicitly, to avoid confusion.*

Thanks - we have explicitly stated simulated particles throughout this paragraph to be clear.

*Line 298-302, figure 9: Is it justified to average trajectories of respectively high- and low N20 events of the entire sample period of one year, given the seasonal variations? E.g. are the trajectories yielding low N2O during winter, similar to those during summer etc.*

Given that the anomalous 'high' and 'low' N20 events that we average over are evenly distributed throughout the year (Fig. 7a), the differences between Fig. 9a and Fig. 9b do not result from seasonal differences. It may be true that some seasonally dependent features are averaged out (for example, a summer anomalous high aerosol case originating from further south versus a winter anomalous high aerosol case originating more locally). However, the purpose of this figure is not to locate exactly where the particles come from geographically during low and high aerosol events (which we would expect to vary seasonally and synoptically), but to highlight the contrast between the two transport pathways: Descent down to the ice sheet from the upper troposphere (associated with high aerosol loading) versus ascent up to Summit from the Ice Sheet margin and surrounding area (associated with low aerosol loading). These features are consistent across the majority of events, independent of seasonality and apparent in the averaged values plotted in Fig. 9. To demonstrate this, we have included thumbnail images of flexpart back trajectories for each individual 'low' and 'high' event in the attached supplement (flexpart_thumbnails.pdf).

*Figure 11: The legend of figure 11a states SBI > 2 °C, while the figure text states SBI > 3 °C. Also > 3 °C is stated earlier in the text. Likely a typo.*
*Figure 13: Same as figure 11*

Good spot, the correct threshold is 3 ◦C (> 0.23 ◦C m−1). We have corrected this in Fig. 11 and Fig. 13.

*Lines 401-403: While I agree on the general concept, that the observed decrease in particle number concentration during fog closely reflects the number concentration of fog droplets, additional particles could be lost to scavenging onto the droplets. Also the statement is not logically sound, as it combines an absolute statement "equal" with a more relative "not necessarily". If something is equal to something, then it absolutely, and not only eventually, excludes anything else.*

We have rephrased this sentence to make it clearer:

**Lines 399 to 402:** *"Importantly, the observed decrease in aerosol particle number concentration during fog events reflects the number of particles that are incorporated into fog droplets too large to pass through the CPC inlet, either through CCN activation and growth or scavenging by fog droplets, and these particles are not necessarily deposited at the surface."*

*Lines 521-523: This is somewhat confusingly written, can this be stated simpler and clearer?*

This paragraph has been removed in the consolidation of the discussion section.

*Line 524: "begin" is written in present tense, should be past tense*

Thanks - we have corrected this.

*Line 526: "reduction in the strength" could be substituted with "weakening"*

We have made this substitution.